# Censor Dependent Variational Inference

**Chuanhui Liu**[1]  **Xiao Wang**[1]

## Abstract

This paper provides a comprehensive analysis of variational inference in latent variable models for survival analysis, emphasizing the distinctive challenges associated with applying variational methods to survival data. We identify a critical weakness in the existing methodology, demonstrating how a poorly designed variational distribution may hinder the objective of survival analysis tasks—modeling time-to-event distributions. We prove that the optimal variational distribution, which perfectly bounds the log-likelihood, may depend on the censoring mechanism. To address this issue, we propose censor-dependent variational inference (CDVI), tailored for latent variable models in survival analysis. More practically, we introduce CD-CVAE, a V-structure Variational Autoencoder (VAE) designed for the scalable implementation of CDVI. Further discussion extends some existing theories and training techniques to survival analysis. Extensive experiments validate our analysis and demonstrate significant improvements in the estimation of individual survival distributions. Codes can be found at https://github.com/ChuanhuiLiu/CDVI.

## 1. Introduction

Survival analysis, a fundamental topic in statistics, finds wide-ranging applications across healthcare, insurance, quality management, and finance (Harrell et al., 2001; Nelson, 2005; Frees, 2009). It focuses on modeling the relationship between time-to-event outcomes and individual demographic covariates, where the event of interest could be death, disease progression, or similar occurrences. A key challenge in survival analysis arises from censored observations, which provide only partial information about the survival time, necessitating specialized methods to handle such data effectively.

Deep learning has emerged as a powerful paradigm to advance survival analysis (Wiegrebe et al., 2024). Recent studies focus on modeling the individual time-to-event distributions using expressive *latent variable survival models* (LVSMs). For example, Ranganath et al. (2016) assumed that the prior of $Z$ belongs to the class of deep exponential family distributions (Brown, 1986). Deep survival machine (Nagpal et al., 2021a) considered the finite discrete latent space, modeling the logarithm of time-to-event as the finite Gumbel or normal mixtures. Xiu et al. (2020) modeled a discrete time-to-event as a softmax-activated neural network incorporating the Nelson-Aalen estimator (Aalen, 1978). Apelláníz et al. (2024) followed a similar setup, developing variational autoencoders (Kingma & Welling, 2014; Rezende et al., 2014) (VAE) for continuous time-to-event. Importantly, these models outperform Accelerated Failure Time (AFT) (Miller, 1976) and Cox Proportional Hazards (CoxPH) (Cox, 1972) models across multiple metrics, enabling downstream tasks through latent features (Manduchi et al., 2022).

A unique aspect of LVSM optimization is its reliance on *variational inference* (VI) (Jordan et al., 1999) to approximate intractable likelihoods arising from flexible parametric generative models. As a result, the performance of latent variable models depends heavily on the optimality of VI and associated design choices (Cremer et al., 2018).

In our opinion, the role of VI in modeling time-to-event distributions remains inadequately explored. It is unclear what form the optimal variational distribution should take under censoring or which posterior it is meant to approximate. Existing designs are largely heuristic, with limited theoretical grounding in the presence of censoring.

To address these gaps, this paper advances the understanding of VI in survival analysis by providing a theoretical analysis of its optimality under censoring and introducing a principled variational framework for LVSM. At a high level, our main contributions are

1. In Section 3, we motivate and introduce the core ideas of *censor-dependent variational inference* (CDVI). We establish conditions under which vanilla VI for survival models fails to achieve inference optimality under non-informative

[1]Department of Statistics, Purdue University, USA. Correspondence to: Xiao Wang <wangxiao@purdue.edu>.

*Proceedings of the 42nd International Conference on Machine Learning*, Vancouver, Canada. PMLR 267, 2025. Copyright 2025 by the author(s).

censoring and expose key limitations of existing methods. We derive the optimal variational posterior for the log-likelihood of survival data. We further demonstrate that a censor-dependent variational family is required to capture the structure of the true posterior, as illustrated in Figure 2.

2. In Section 4, we propose CD-CVAEs, amortized CDVI implementations of VAE-based survival models, free from the restrictive proportional hazards assumption and imposing minimal distributional assumptions, which rely solely on decoder architecture. We provide formal guarantees for our proposed tighter-bound sampling-based variants via augmented VI theory and offer practical insights into model implementations, such as training strategies and stable computation techniques. Section 5 presents empirical validation and findings through extensive experiments on various datasets. Notably, our methods achieve 5% higher C-index than state-of-the-art models on WHAS datasets.

## 2. Preliminaries

**Notations**: Random variables (r.v.) are denoted by capital letters, e.g. $X, Z, Y, U, C$, and their distribution functions have matching subscripts. $\mathcal{X}$ denotes the sample space of $X$. $P(\cdot), F(\cdot), p(\cdot), S(\cdot), h(\cdot)$ respectively denote a general probability function, a cumulative distribution function, a density function, a survival (tail) function, and a hazard function. Subscripts in Greek letters $\theta, \phi$ denote the unknown parameters. E.g. $S_{Y,\theta}(\cdot)$ refers to the survival function of $Y$ parameterized by $\theta$. Additional letters $f, q$ denote different density functions, e.g., $f_\theta(\cdot) = p_{U,\theta}(\cdot), q_\phi(\cdot) = p_{Z,\phi}(\cdot)$. A proportional relationship over $x$ is denoted as $\propto_x$. Estimates of functions or random variables are indicated with a caret or dot symbol above, e.g., $\hat{S}(\cdot)$ is an estimate of $S(\cdot)$. log denotes natural logarithms. Bold symbol $\boldsymbol{x}$ denotes vectors.

### 2.1. Log-likelihood for Right-censored Data

**Datasets**: In a single-event right-censoring setting, we consider a non-longitudinal survival dataset consisting of $n$ triplets $\{\boldsymbol{x}_i, y_i, \delta_i\}_{i=1}^n$. In particular, the event indicator $\delta_i = 1$ signifies that $y_i$ is the observed time of the event of interest (time-to-event), while $\delta_i = 0$ signifies that $y_i$ is right-censored and the true time-to-event of subject $i$ with individual features/covariates $x_i$ exceeds the observed value.

In this work, we assume the dataset consists of i.i.d. random variables $\{X, Y, I\}$, where observed survival time $Y$ is continuous. Notably, we consider $(Y, I)$ as the surjective transformation of two continuous latent variables under the independent censoring assumption $U \perp\!\!\!\perp C \mid X$:

$$Y = \min(U, C), \quad I = \mathbb{1}(U \le C), \tag{1}$$

where $U$ is the uncensored time-to-event and $C$ is the censoring time. Here, we denote by $\theta, \eta$ the unknown pa-

rameters governing the distribution of $U, C$, respectively.

The goal of time-to-event modeling is to estimate the event-time distribution parameterized by $\theta$. Under such censoring assumptions, the *joint density*[1] of $y, \delta$ conditioned on $\boldsymbol{x}$ can factorize as follows

$$p_{\theta,\eta}(y, \delta|\boldsymbol{x}) \propto_\theta p_{U,\theta}(y|\boldsymbol{x})^\delta S_{U,\theta}(y|\boldsymbol{x})^{1-\delta}. \tag{2}$$

Taking the log, the right-hand side of (2) gives rise to the full-parametric objective $L(\theta)$, which is given by

$$L(\theta) := \delta \log f_\theta(y|\boldsymbol{x}) + (1 - \delta) \log S_\theta(y|\boldsymbol{x}), \tag{3}$$

where $f_\theta(y|\boldsymbol{x}), S_\theta(y|\boldsymbol{x})$ represents the density, survival functions of $U$ evaluated at observed time $y$, respectively. While (3) is called the log-likelihood parameterized by $\theta$ (Kalbfleisch & Prentice, 2002), we note that the right-hand side of (2) is not a proper density, as it lacks a normalizing constant with respect to $\eta$.

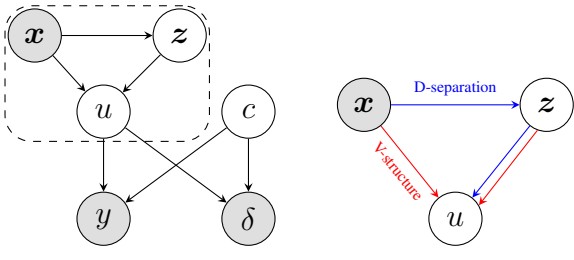

(a) generative graph of LVSM     (b) latent structures

Figure 1: Directed acyclic graphs of LVSM. The shaded nodes $\boldsymbol{x}, y, \delta$ are observed. (a) The dashed box shows a general generative graph of $U$. (b) Specific Latent structures: D-separation assumes $X \perp\!\!\!\perp U|Z$; V-structure assumes a $X$-independent latent $Z$. Best viewed in color.

### 2.2. Latent Variable Survival Model (LVSM)

LVSMs construct $f_\theta(u|x)$ from (3) within a latent structure using a continuous latent variable $Z$, enabling a more flexible and expressive characterization than traditional methods. As shown in Fig.1, a general formulation is given by

$$f_\theta(u|\boldsymbol{x}) = \int_{z \in \mathcal{Z}} f_\theta(u|\boldsymbol{x}, \boldsymbol{z}) p_\theta(\boldsymbol{z}|\boldsymbol{x}) dz, \tag{4}$$

where $p_\theta(\boldsymbol{z}|\boldsymbol{x})$ is the conditional prior of $Z$ and $f_\theta(u|\boldsymbol{x}, \boldsymbol{z})$ is often called the decoder or the emission distribution.

Such latent structure enhances the expressiveness of survival models $f_\theta$, moving beyond the constraints of proportional hazards assumptions. For example, an AFT model can be seen as LVSM constrained by a linear latent, e.g., $Z|X = \alpha + \beta^\top X$, in a D-separation structure illustrated in Fig 1b.

---

[1]Radon–Nikodym derivative of the distribution $P(Y, I|X)$ w.r.t. the product of the Lebesgue and counting measure.

While LVSM is more flexible, the M-estimation of $\theta$, i.e., $\hat{\theta}_{mle} = \arg\max L(\theta)$ is challenging due to its computational cost. Specifically, $f_\theta$ in (3) may lack a closed-form integral, rendering it even harder to approximate $S_\theta$ reliably. As a solution, VI is one common techniques used in LVSM.

## 2.3. Vanilla Variational Inference for LVSM

Here, we review a naive framework of VI, referred to as the **vanilla VI**, as seen in Ranganath et al. (2016); Xiu et al. (2020); Nagpal et al. (2021b); Apellániz et al. (2024). Specifically, a variational distribution $q_\phi(z|x, y)$ is proposed to obtain unbiased estimators of $f_\theta(y|x), S_\theta(y|x)$. By Jensen's inequality, $\log f_\theta(y|x), \log S_\theta(y|x)$ in (3) can therefore be lower bounded by

$$\log f_\theta(y|x) \geq \mathbb{E}_{q_\phi} \log f_\theta(y|x, z) - \mathrm{KL}[q_\phi||p_\theta(z|x)], \quad (5)$$

$$\log S_\theta(y|x) \geq \mathbb{E}_{q_\phi} \log S_\theta(y|x, z) - \mathrm{KL}[q_\phi||p_\theta(z|x)]. \quad (6)$$

By substituting the right-hand sides of (5) and (6) in (3), it yields a lower bound $\mathrm{ELBO}(\theta, \phi)$ on the log-likelihood (Xiu et al., 2020; Nagpal et al., 2021a), which is given by

$$\begin{aligned} \mathrm{ELBO}(\theta, \phi) &:= \delta \mathbb{E}_{q_\phi} \log f_\theta(y|x, z) \\ &+ (1 - \delta) \mathbb{E}_{q_\phi} \log S_\theta(y|x, z) - \mathrm{KL}[q_\phi||p_\theta(z|x)]. \end{aligned} \quad (7)$$

As a side note, Ranganath et al. (2016); Apellániz et al. (2024) further decompose the $\mathrm{KL}[q_\phi||p_\theta(z|x)]$ (KLD) without assuming that $p_\theta(z|x)$ is tractable, where intractable $\log p_\theta(x)$ is moved into $L(\theta)$ by rearrangement.

$$\mathrm{KLD} = \log p_\theta(x) + \mathrm{KL}[q_\phi||p(z)] - \mathbb{E}_{q_\phi} \log p_\theta(x|z). \quad (8)$$

Consequently, (7) enables tractable and efficient computation of both the expectation and the KL divergence, improving scalability for large datasets. Often, optimizing (7) can be done by amortized black-box VI algorithms (Ranganath et al., 2014) via the reparameterization trick (Kingma & Welling, 2014; Rezende et al., 2014).

## 2.4. Inference Optimality for Censored Data

The key distinction in optimizing (7), rather than directly maximizing $L(\theta)$, lies in its pursuit of two distinct objectives *simultaneously*: 1) the M-estimation of $\theta$ and 2) the variational bound of $L(\theta)$ defined in (3). The second objective aims to minimize the *inference gap* (Cremer et al., 2018), i.e., the bias, of $L(\theta)$:

$$B(\theta, \phi) := L(\theta) - \mathrm{ELBO}(\theta, \phi). \quad (9)$$

Since the optimum $(\theta^*, \phi^*) := \arg\max \mathrm{ELBO}(\theta, \phi)$ balances the best of these two results, the optimality of VI is crucial for the estimation accuracy of $\theta^*$. Improper variational approximations introduce bias and degrade the reliability of estimates of $\theta$.

Existing approaches of variational conditional posterior fail to extend to survival analysis. In a (semi-)supervised setting (Kingma et al., 2014; Sohn et al., 2015), VI aims to approximate the intractable posterior $p_\theta(z|x, y)$ with $q_\phi(z|x, y)$. However, these methods struggle to account for censoring in the observed labels, i.e., the time-to-event $U$ in our case.

On the other hand, existing methods of LVSMs often lack rigorous inference optimality analysis, and their design can appear counter-intuitive from a Bayesian perspective. For example, Nagpal et al. (2021a) adopted a lazy strategy in obtaining $q_\phi$ by manually setting $q_\phi$ equal to the tractable $p_\theta(z|x)$. Similarly, Apellániz et al. (2024) limited $q_\phi$ to depend on $X$ only, making it completely ignore the information of $y$.

# 3. Theories

This section studies inference optimality in LVSMs, where optimality of $\phi$ is defined as tightly bounding the log-likelihood (3), parameterized by $\theta$, assuming at least one censored and one uncensored observation.

## 3.1. Problems in vanilla VI

### 3.1.1. AN EDGE CASE STUDY

As a preliminary step toward understanding the problems of vanilla VI, we start by analyzing the equality (tightness) conditions of log-likelihood components in (5) and (6) as functions of $x, u$.

**Lemma 3.1** (**Equality conditions of** (5) **and** (6)).
*Given any parameter $\theta$, the point-wise equality in (5) holds for any $\{X = x, U = u\}$, if and only if one of the following conditions holds:*

(a) $q_\phi(z|x, u) = f_\theta(u, z|x)/f_\theta(u|x)$, where $f_\theta(u, z|x) := f_\theta(u|x, z)p_\theta(x|z)$;
(b) $\exists$ map $c_1$, $f_\theta(u, z|x)/q_\phi(z|x, u) = c_1(x, u)$;
(c) $\mathrm{KL}[q_\phi(z|x, u)||p_\theta(z|x, u)] = 0$.

*Likewise, (6) holds for any $\{X = x, U = u\}$, if and only if the following equivalent conditions hold:*

(a') $q_\phi(z|x, u) = S_\theta(u, z|x)/S_\theta(u|x)$, where we abuse $S_\theta(u, z|x) := \int_{s=u}^\infty f_\theta(s, z|x)ds$;
(b') $\exists$ map $c_2$, $S_\theta(u, z|x)/q_\phi(z|x, u) = c_2(x, u)$.

Proof: the conditions for (5) follow the standard VI argument, and the conditions for (6) are derived under the additional assumption of Fubini's Theorem.

Lemma 3.1 raises a natural question about the variational inference optimality in survival analysis:

*Given any $\theta$ and $(x, u)$, what kind of $q_\phi(z|x, u)$ would satisfy both tightness conditions ?*

In other words, we ask: *is there an optimal variational solution that have a zero inference gap of log-likelihood given both data $\{\boldsymbol{x}, y, 0\}$ and $\{\boldsymbol{x}, y, 1\}$?*

For notation clarity, let $\Phi_1(\theta)$ denote the set of $\phi$ where (5) holds equal, $\Phi_2(\theta)$ denote the one for (6). Thus, we are interested in its union $\Phi_U(\theta) := \Phi_1(\theta) \cap \Phi_2(\theta)$, which is the ideal parameter set for optimal $q_\phi$ that eliminates the bias of (7) and has a zero inference gap. On top of that, we define $\Theta_U := \{\theta \mid \Phi_U(\theta) \neq \varnothing\}$ to denote the support set of $\theta$ allowing optimal vanilla VI.

Perhaps surprisingly, Proposition 3.1 below highlights the degradation of optimal vanilla VI solution, showing that these conditions in Lemma 3.1 are fundamentally different.

**Proposition 3.1** (**Degradation for optimal** $q_\phi(\boldsymbol{z}|\boldsymbol{x}, u)$)**.**
*Assuming that 1) optimal VI is feasible: $\Theta_U \neq \varnothing$, and 2) $f_\theta(u|\boldsymbol{x}, \boldsymbol{z})$ is a location-scale density with location $\mu_\theta(\boldsymbol{x}, \boldsymbol{z})$ and scale $\sigma$. Then, given any $x, u$,*

(1) *Latent non-identifiability: $\forall \theta \in \Theta_U$, hazard function $h_\theta(u|\boldsymbol{z}, \boldsymbol{x})$ is independent of $\boldsymbol{z}$;*
(2) *Location degradation: $\forall \theta \in \Theta_U$, location parameter $\mu_\theta(\boldsymbol{x}, \boldsymbol{z})$ is independent of $\boldsymbol{z}$;*
(3) *Lazy posterior: $\forall \theta \in \Theta_U$, $\forall \phi \in \Phi_U(\theta)$, the variational distribution $\mathrm{KL}[q_\phi(\boldsymbol{z}|\boldsymbol{x}, u) \| p_\theta(\boldsymbol{z}|\boldsymbol{x})] = 0$;*
(4) *Surely posterior collapse: If $\boldsymbol{z} \perp\!\!\!\perp \boldsymbol{x}$, $\forall \theta \in \Theta_U, \phi \in \Phi_U(\theta)$, $\mathrm{KL}[q_\phi(\boldsymbol{z}|\boldsymbol{x}, u) \| p(\boldsymbol{z})] = 0$.*

Proofs are deferred to Appendix B.1. In a nutshell, claims (1) and (2) reveal the critical limitation that optimal VI can only be achieved on extremely limited support of $\theta$. Specifically, claim (1) asserts that $h_\theta(u|\boldsymbol{x}, \boldsymbol{z})$, or equivalently $f_\theta(u|\boldsymbol{x}, \boldsymbol{z})$, is independent of $\boldsymbol{z}$, disregarding the latent information from prior $p_\theta(\boldsymbol{z}|\boldsymbol{x})$. Remarkably, such behavior of $f_\theta(u|\boldsymbol{x}, \boldsymbol{z})$, known as *latent non-identifiability* (Wang et al., 2021), is first identified in the context of survival analysis. Under additional location-scale distribution assumption, claim (2) asserts that its mean $\mu_\theta(\boldsymbol{x}, \boldsymbol{z})$ reduces to an univariate function that is independent of $\boldsymbol{z}$, restricting the expressiveness of LVSM. This observation may help explain why the aforementioned applications often adopt a D-separated latent structure, where $f_\theta(u|\boldsymbol{x}, \boldsymbol{z})$ is fully dependent on $z$, in an effort to mitigate or avoid such expressiveness issues.

Moreover, claims (3) and (4) demonstrate the negligibility of the optimal solution $q_\phi$. The reason is simple—since both $f_\theta(u|\boldsymbol{x}, \boldsymbol{z})$ and $S_\theta(u|\boldsymbol{x}, \boldsymbol{z})$ are independent of $\boldsymbol{z}$, their posterior equals nothing but their common prior. claim (3) proves that the optimal $q_\phi$ collapses to the conditional prior $p_\theta(\boldsymbol{z}|\boldsymbol{x})$, ignoring the information of $u$. To this extent, the optimal $q_\phi$ becomes as lazy as the one proposed in Nagpal et al. (2021a). It also explains the rationale in Apellániz et al. (2024), where the proposed $q(\boldsymbol{z}|\boldsymbol{x})$ is not dependent on $u$. Such negligibility can be more detrimental if the latent is V-

structured, e.g., the latent $\boldsymbol{z}$ represents an unseen individual-independent treatment. Claim (4) states that optimal $q_\phi$ is collapsed to the prior $p(\boldsymbol{z})$, which leads to a notorious issue called posterior collapse (Bowman et al., 2016; Lucas et al., 2019).

### 3.1.2. ON CENSORING ASSUMPTIONS

We next extend the optimality analysis by considering when $q_\phi(\boldsymbol{z}|\boldsymbol{x}, u)$ must simultaneously satisfy both inequalities in (5) and (6). As previously discussed, this situation arises when both data points $\{\boldsymbol{x}, y, 0\}$ and $\{\boldsymbol{x}, y, 1\}$ are present in the datasets. In the infinite data limit, it becomes essential to examine the underlying sampling space, which defines the supports of (5) and (6). For clarity, we denote the event space $\mathcal{D}_E$ and the censored space $\mathcal{D}_C$ as follows,

$$
\begin{aligned}
\mathcal{D}_E &:= \{(\boldsymbol{x}, y) \mid (\boldsymbol{x}, y, 1) \in \mathcal{X} \times \mathcal{Y} \times \mathcal{I}\}, \\
\mathcal{D}_C &:= \{(\boldsymbol{x}, y) \mid (\boldsymbol{x}, y, 0) \in \mathcal{X} \times \mathcal{Y} \times \mathcal{I}\}.
\end{aligned}
\tag{10}
$$

Remark 3.1 below delineates the conditions under which the issues in Proposition 3.1 persist in the infinite data limit.

**Remark 3.1.** For any $(\boldsymbol{x}, y) \in \mathcal{D}_E \cap \mathcal{D}_C$, Proposition 3.1 is applicable to the optimal $q_\phi(\boldsymbol{z}|\boldsymbol{x}, y)$.

Specifically, if $\mathcal{D}_E \cap \mathcal{D}_C \neq \varnothing$, such issues of vanilla VI are unavoidable. On the other hand, if $\mathcal{D}_E \cap \mathcal{D}_C = \varnothing$, it is *theoretically* possible for vanilla VI to achieve a zero inference gap, satisfying the conditions of (5) on $\mathcal{D}_E$ and (6) on $\mathcal{D}_C$. In other words, the type of censoring and, more importantly, its effect on the partition of the sample space are crucial to the vanilla VI optimality.

Here, we note that the widely adopted non-informative censoring assumption (Lagakos, 1979) lacks specificity on vanilla VI optimality, since its influence on the partitioning of the sample space is not explicitly characterized. For example, under certain types of non-informative censoring, such as Type-I censoring (Lawless, 2003), it is possible that the event and censoring spaces become disjoint. That said, evident in benchmark datasets (See Table 3), observational studies rarely have disjoint spaces; vanilla VI can be suboptimal in these benchmark datasets.

### 3.2. Censor-dependent Variation Inference (CDVI)

Having identified the limitations of vanilla VI, we are now ready to establish a less restrictive VI framework for LVSM.

#### 3.2.1. OPTIMAL VI FOR JOINT DENSITY

First, we revisit the setup and establish the optimal variational distribution for the joint density in (2).

**Theorem 3.2.1** (**Point-wise optimal VI**)**.**
*Given $\boldsymbol{x}, y, \delta$ and parameter $\theta$, $q_\phi(\boldsymbol{z}|\boldsymbol{x}, y, \delta)$ is optimal if*

*and only if for almost every $\boldsymbol{z} \in \mathcal{Z}$,*

$$q_{\phi^*}(\boldsymbol{z}|\boldsymbol{x}, y, \delta) = \lim_{vol(\Delta \boldsymbol{z}) \to 0} \frac{P_{\theta, \eta}(\boldsymbol{z} \leq Z \leq \boldsymbol{z} + \Delta \boldsymbol{z}|\boldsymbol{x}, y, \delta)}{vol(\Delta \boldsymbol{z})}.$$

*Moreover, if $\mathcal{D}_E = \mathcal{D}_C = \mathcal{X} \times \mathcal{U}$, the optimal $q_{\phi^*}$ is independent of parameters of the censoring distribution $\eta$, and for almost every $\boldsymbol{z} \in \mathcal{Z}$,*

*(a) $q_{\phi^*}(\boldsymbol{z}|\boldsymbol{x}, y, 1) = q_{\phi_1^*}(z|x, u)|_{u=y}$, where $\phi_1^* \in \Phi_1(\theta)$.*
*(b) $q_{\phi^*}(\boldsymbol{z}|\boldsymbol{x}, y, 0) = q_{\phi_2^*}(z|x, u)|_{u=y}$, where $\phi_2^* \in \Phi_2(\theta)$.*

The proof is deferred in Appendix B.2. Thm 3.2.1 states that the optimal variational distribution $q_\phi$ for joint density (2) is equal to the posterior density of $P(Z|X, Y, \delta)$. In particular, if there is no overlap of sample spaces due to censoring, the optimal $q_\phi$ becomes independent of $\eta$ and thus becomes the one for log-likelihood satisfying Lemma 3.1.

Thm 3.2.1 presents an alternative perspective on the source of the previously identified limitation, attributing it to the design $q_\phi$ in vanilla VI.

**Remark 3.2 (Vanilla VI propose a marginal $q_\phi$).**
Assuming that there is no partition $\mathcal{D}_E = \mathcal{D}_C = \mathcal{X} \times \mathcal{U}$, the marginalized $q_{\phi^*}(\boldsymbol{z}|\boldsymbol{x}, y)$ equals $q_{\phi_i^*}(\boldsymbol{z}|X = \boldsymbol{x}, U = y)$ for any $i = 1, 2$ if and only if $P(\delta = 2 - i|Y = y) = 1$.

Remark 3.2 states that the inability to obtain equality in both (5) and (6) arises from defining $q_\phi$ as a *marginal* distribution, whereas the true posterior it aims to approximate is a conditional one. As illustrated in Fig.2, $q(\boldsymbol{z}|\boldsymbol{x}, y)$ proposed by vanilla VI is $\delta$-marginalized and lacks the censor-dependent structure. Consequently, the optimal solution of vanilla VI would approximate the true posterior if and only if there is no event or censoring data. From this point, further limitations on $q_\phi$ described in Section 2.3, such as employing a lazy strategy or making it independent of $y$, are irrational.

### 3.2.2. CENSOR-DEPENDENT ELBO

$q_\phi(\boldsymbol{z}|\boldsymbol{x}, y, \delta)$ in Thm 3.2.1 is called the *censor-dependent* due to the explicit dependency on the indicator $\delta$. To derive the corresponding ELBO, we now introduce an equivalent parameterization.

**Definition 3.2 (Alternative Parameterization of $q_\phi$).**
*The censor-dependent variational distribution $q_{\phi_1, \phi_2}$ can be parameterized as*

$$q_{\phi_1, \phi_2}(\boldsymbol{z}|\boldsymbol{x}, y, \delta) := q_{\phi_1}(\boldsymbol{z}|\boldsymbol{x}, y)^\delta q_{\phi_2}(\boldsymbol{z}|\boldsymbol{x}, y)^{1-\delta}. \quad (11)$$

It is important to note that, under this parameterization, the optimal $\phi_1$ and $\phi_2$ depend on the value of $\theta$ and are not mutually independent. See Thm.3.2.2 for more discussion. Consequently, in practice, either $\phi_1$ or $\phi_2$ is a partition of network parameters, which are not trained separately on all event or censored observations. This choice of parameterization improves clarity in the following formulations.

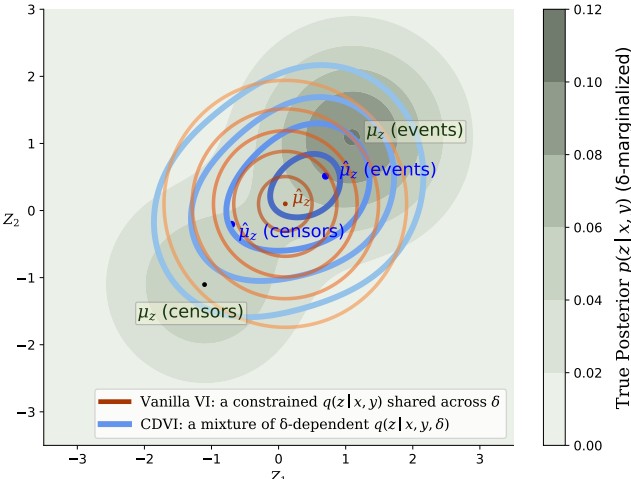

Figure 2: Comparison on simulated datasets $SD4$ (see Table 2). The learned posterior is marginalized over censoring status $\delta$, conditioned on $x = 1, y = 0$. The closed-form true posterior is given in Appendix C.2. Best viewed in Color.

Specifically, it defines the induced likelihood estimators :

$$\begin{aligned} \hat{f}_1(\boldsymbol{z}) &:= f_\theta(y, \boldsymbol{z}|\boldsymbol{x})/q_{\phi_1}(\boldsymbol{z}|\boldsymbol{x}, y), \\ \hat{S}_1(\boldsymbol{z}) &:= S_\theta(y, \boldsymbol{z}|\boldsymbol{x})/q_{\phi_2}(\boldsymbol{z}|\boldsymbol{x}, y). \end{aligned} \quad (12)$$

The subscript 1 is purposefully added to align the notation in sampling variants defined in (14). Plugging in (12) in (3) and taking expectation, we obtain the **Censor-dependent ELBO** (ELBO-C):

$$\begin{aligned} \text{ELBO-C} := &\delta[\mathbb{E}_{q_{\phi_1}} \log f_\theta(y|\boldsymbol{x}, \boldsymbol{z}) - \text{KL}[q_{\phi_1} \| p_\theta(\boldsymbol{z}|\boldsymbol{x})]] \\ &+ (1 - \delta)[\mathbb{E}_{q_{\phi_2}}[\log S_\theta(y|\boldsymbol{x}, \boldsymbol{z})] - \text{KL}[q_{\phi_2} \| p_\theta(\boldsymbol{z}|\boldsymbol{x})]]. \end{aligned} \quad (13)$$

Compared to vanilla ELBO (7), we demonstrate its suitability for survival analysis below, along with how it addresses the previously identified issues. Let $\Phi_P(\theta) = \{(\phi_1, \phi_2) \mid \phi_1 \in \Phi_1(\theta), \phi_2 \in \Phi_2(\theta)\}$ denote the set of optimal parameters of CDVI.

**Theorem 3.2.2 (Informal; CDVI optimality).**
*If $\Phi_P(\theta) \neq \varnothing$, $\forall(\phi_1, \phi_2) \in \Phi_P(\theta)$, $q_{\phi_1, \phi_2}(\boldsymbol{z}|\boldsymbol{x}, y, 0) \propto_z h_\theta(y|\boldsymbol{x}, \boldsymbol{z})q_{\phi_1, \phi_2}(\boldsymbol{z}|\boldsymbol{x}, y, 1)$ and $q_{\phi_1, \phi_2}$ do not have issues in proposition 3.1 on a larger support of $\theta$.*

The complete formal version is included in Appendix A.2 and its proof can be found in Appendix B.3. In short, Thm 3.2.2 highlights that the censor-dependent structure in $q_{\phi_1, \phi_2}$ eliminates the problematic constraint $\phi_1 = \phi_2$ in vanilla VI.

To conclude, we have shown that the vanilla VI framework described in Ranganath et al. (2016); Xiu et al. (2020); Nagpal et al. (2021a); Apellániz et al. (2024) is insufficient and arguably inappropriate for LVSM. Without hindering the

M-estimation of $\theta$ and the expressiveness of latent survival models, we have shown the importance of the censoring mechanism and have introduced the censor-dependent structure for optimal VI in LVSM.

## 4. Methods

In this section, we discuss the implementations of amortized CDVI in VAE-based LVSMs, sharing insights into its optimization and CDVI augmentation techniques.

### 4.1. Censor-dependent Conditional VAE

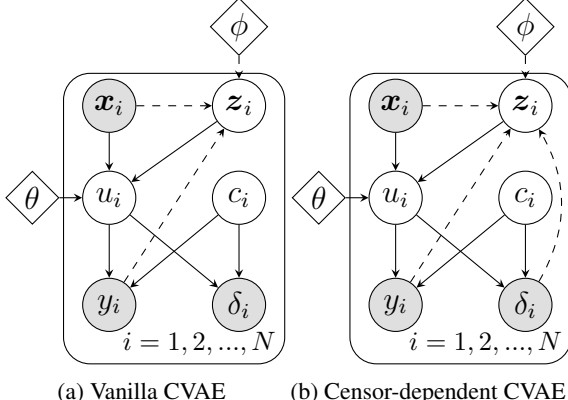

(a) Vanilla CVAE  (b) Censor-dependent CVAE

Figure 3: Implementations of Vanilla VI and CDVI.

First, we propose the Censor-dependent Conditional VAE (CD-CVAE) that estimates parameters $\theta, \phi$ as weights of neural networks. As illustrated in Fig.3, our proposed CDVI implementation uniquely incorporates both $y$ and the event indicator $\delta$ as input of the encoder.

Fig.4 illustrates the structure of the encoder and decoder. Our proposed decoder adopts a novel latent V-structure and employs both *Gaussian* and *Gumbel-minimum* (Gumbel, 1958) distribution families for $\varepsilon$, which can be interpreted as an infinite LogNormal or Weibull mixture survival regression on positive survival time. We note that the decoder parameter $\theta$ is decomposed as $\{\zeta, \sigma\}$ in Fig.4b.

### 4.2. Training Strategy of Decoder Variance

Censored labels limit the flexibility of CD-CVAE's training procedure, restricting the use of advanced strategies available to standard CVAE models. As emphasized in Prop. 4.2, a dual-step algorithm that updates $\sigma$ separately, as seen in Rybkin et al. (2021) and Liu & Wang (2025), is not applicable, since $\sigma$ cannot be estimated in closed form.

**Proposition 4.2** (**No closed form update of** $\sigma$). *Given the dataset* $\{\boldsymbol{x}_i, y_i, \delta_i\}_{i=1}^n$ *and decoder mean* $\zeta$, *encoder parameters* $\phi_1, \phi_2$, *the optimum of* $\sigma$ *by* $\frac{\partial \text{ELBO-C}(\phi,\zeta,\sigma)}{\partial \sigma} = 0$ *has no closed-form solution. In particular, if* $\varepsilon$ *follows a*

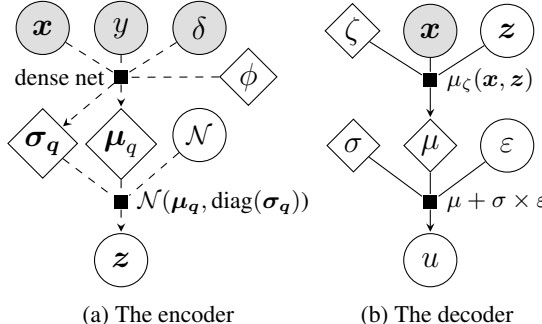

(a) The encoder  (b) The decoder

Figure 4: Generative graph of CD-CVAE.

*standard normal distribution, we have*

$$\frac{\partial \text{ELBO-C}}{\partial \sigma} = \mathbb{E}_q \big[ \sum_{i:\delta_i=1} \big( \frac{\tilde{y}_i^2}{\sigma} - \frac{1}{\sigma} \big) + \sum_{i:\delta_i=0} h(\tilde{y}_i) \frac{\tilde{y}_i}{\sigma} \big],$$

*where* $\tilde{y} = (y - \mu_\zeta(\boldsymbol{x}, \boldsymbol{z}))/\sigma$ *is the standardized* $y$ *and* $h()$ *is standard normal hazard function.*

The proof of Prop.4.2 is given in Appendix B.4.

### 4.3. Augmented CDVI and the Implementations

Next, we introduce two variants of our proposed model, namely IS and DVI, incorporating established augmented VI techniques. We formulate the corresponding log-likelihood estimators so that the corresponding ELBO can be easily obtained as forms of their expectations.

**Definition 4.3.1** (The Importance Sampling Variant (IS)). *Following Definition 3.2, the unbiased Monte Carlo estimators of likelihood* $f_\theta(y|x), S_\theta(y|x)$ *are defined as*

$$\hat{f}_m := \frac{1}{m} \sum_{i=1}^m \hat{f}_1(\boldsymbol{z}_i), \ \hat{S}_k := \frac{1}{k} \sum_{j=1}^k \hat{S}_1(\boldsymbol{z}_j), \qquad (14)$$

*where* $\boldsymbol{z}_i$ *and* $\boldsymbol{z}_j$ *are independent samples from* $q_{\phi_1,\phi_2}$, *assuming* $\delta = 1, 0$, *respectively, and* $m$ *and* $k$ *denotes the corresponding sample size.*

Similarly, (14) defines a general $\hat{L}_{m,k} := \log(\hat{f}_m^\delta \hat{S}_k^{1-\delta})$ for $L(\theta)$. Computing its expectation allows us to generalize (13) to ELBO-C$_{(m,k)}$ and (9) to $B(m,k)$ defined below.

For completeness, we establish 3 results about the properties of $\hat{L}_{m,k}$, providing deeper insights into augmented CDVI in both the finite-sample setting and the asymptotic regime as $m, k \to \infty$. See detailed proofs in Appendix B.5 to B.7

**Theorem 4.3.1** (**Monotonicity of Inference Gaps**).
*Given any* $\theta, \phi$, *for any* $m \in \mathbb{N}_+, k \in \mathbb{N}_+$,
$B(1,1) \geq B(m,k) := L(\theta) - \text{ELBO-C}_{m,k}(\theta, \phi)$
$\qquad \geq \max(B(m, k+1), B(m+1, k))$
$\qquad \geq B(m+1, k+1) \geq \lim_{m',k' \to \infty} B(m', k') = 0$

The dependency of $B(m, k)$ on model parameters $\theta, \phi$ is omitted; $B(1, 1)$ is equal to the gap of (13) in Thm 4.3.1.

Thm 4.3.1 generalizes the well-known property of Burda et al. (2015) to labeled datasets with censoring. We prove that the generalized inference gap $B(m, k)$ is monotonic in both size $m$ and $k$. In other words, ELBO-C$_{m,k}$ yields a smaller inference gap for any $m > 1, k > 1$ given a *fixed* $\theta, \phi_1, \phi_2$, which vanishes as $m, k \to \infty$. That said, Thm 4.3.1 holds for any choice of $\phi_1, \phi_2$, including the case $\phi_1 = \phi_2$ imposed by vanilla VI, as in Xiu et al. (2020).

**Theorem 4.3.2 (Self-normalized Importance Sampling).** *Let $Q_1(m), Q_2(k)$ be the augmented variational distribution, and $P_1(m), P_2(k)$ be the augmented posterior distribution, defined as follows:*

$$
\begin{aligned}
&J_1(m) = \hat{f}_1 \prod_{i=1}^{m} q_{\phi_1}(\boldsymbol{z}_i|\boldsymbol{x}, y), \; Q_1(m) = J_1(m)/\hat{f}_m \\
&J_2(k) = \hat{S}_1 \prod_{j=1}^{k} q_{\phi_2}(\boldsymbol{z}_j|\boldsymbol{x}, y), \; Q_2(k) = J_2(k)/\hat{S}_k \\
&P_1(m) \propto_{z_{1:m}} J_1(m), \; P_2(k) \; \propto_{z_{1:k}} J_2(m).
\end{aligned}
\tag{15}
$$

*Then, given any $x, y, \delta$,*

$$
\begin{aligned}
&L(\theta) - \mathbb{E}_{Q_1, Q_2}[\hat{L}_{m,k}] \\
&= \mathrm{KL}[Q_1(m)||P_1(m)]^{\delta} \mathrm{KL}[Q_2(k)||P_2(k)]^{(1-\delta)}.
\end{aligned}
\tag{16}
$$

Thm 4.3.2 extended and corrected the results from Domke & Sheldon (2018), formulating the inference gaps of augmented CDVI as the KL divergence. This result generalizes the established connection of self-normalized importance sampling (SNIS) to CDVI. For example, $\hat{f}_1/\hat{f}_m$ acts as a form of self-normalization. However, we note that it does not enable a direct comparison between $B(m, k)$ and $B(1, 1)$, since the expectation is taken over $Q_1$ and $Q_2$. Detailed discussion can be found in Appendix B.6.

**Theorem 4.3.3 (Informal; Consistency of estimators).** *Under some moment assumptions, for $m \to \infty, k \to \infty$, the variance of $\hat{L}_{m,k}$ goes to zero, and thus $\hat{L}_{m,k}$ is a biased yet consistent estimator of $L(\theta)$, i.e., for any $\xi > 0$,*

$$
\lim_{m,k \to \infty} P_z(|\hat{L}_{m,k} - L(\theta)| > \xi) = 0.
$$

A formal version is provided in Appendix A.3. Despite that Thm 4.3.1 has shown a vanishing bias of $\hat{L}_{m,k}$, Thm 4.3.3 quantifies the asymptotic behavior of its variance, thereby establishing its consistency. This result is extended from Nowozin (2018), enhancing CDVI under ideal assumptions with theoretical guarantees. Thm 4.3.3 also enables the following trade-off for a smaller asymptotic bias.

**Definition 4.3.2 (The Delta Method Variant (DVI)).** *Following Definition 3.2, A biased variant of Definition 4.3.1 is defined as*

$$
\dot{f}_m := \exp\{\hat{\alpha}_2/(2m\hat{f}_m^2)\}\hat{f}_m,
\tag{17}
$$

$$
\dot{S}_k := \exp\{\hat{\beta}_2/(2k\hat{S}_k^2)\}\hat{S}_k,
\tag{18}
$$

*where we define $\hat{\alpha}_2$ and $\hat{\beta}_2$ as the corresponding sample variances of $\{\hat{f}_1(\boldsymbol{z}_i)\}_{i=1}^{m}$ and $\{\hat{S}_1(\boldsymbol{z}_i)\}_{i=1}^{k}$, e.g., $\hat{\alpha}_2 := \frac{1}{m-1}\sum_{i=1}^{m}(\hat{f}_1(\boldsymbol{z}_i) - \hat{f}_m)^2$.*

Inspired from Nowozin (2018), we prove in Appendix A.4 that the Delta method (Teh et al., 2006) variant $\dot{L}_{m,k}$ enjoys less asymptotic bias of $L(\theta)$ compared to (14), if $m, k$ are sufficiently large. For practical implementation, the Log-SumExp (Log-Softmax) trick used to compute the bias term stably is detailed in Appendix C.5.

# 5. Experiments

The additional details of experiments are in Appendix C.

## 5.1. Evaluation Metrics

**Concordance index** (Harrell et al., 1982; Uno et al., 2011) measures the effectiveness of a discriminative model in ranking survival times correctly. Specifically, it assesses whether the model assigns a shorter predicted time to the event, or more generally, a lower survival probability $\hat{S}(t|\boldsymbol{x}_i)$ at test time $t$, for a subject with features $\boldsymbol{x}_i$ who experienced the event at time $y_i$, compared to a subject with features $x_j$ who survived longer. Due to censoring, only comparable pairs $y_i \le y_j, \delta_i = 1$ are considered. Formally, the $C$-index is defined as:

$$
C(t) = P(\hat{S}(t|\boldsymbol{x}_i) \le \hat{S}(t|\boldsymbol{x}_j) \mid y_i \le y_j, \delta_i = 1).
$$

We evaluate the trained models by calculating the average C-index over ten quantiles, ranging from $10^{th}$ to $100^{th}$ quantile in increments of 10, of event test times.

**Brier score** (Brier, 1950; Graf et al., 1999) is a squared prediction error reweighted by Inverse Probability of Censoring Weighting (IPCW), designed to assess both calibration and discrimination (Haider et al., 2020).

$$
\begin{aligned}
\mathrm{Brs}(t) = \frac{1}{n} \sum_{i=1}^{n} [&\mathbb{1}(y_i \le t, \delta_i = 1)\frac{(0 - \hat{S}(t|\boldsymbol{x}_i))^2}{\hat{S}_C(y_i)} \\
&+ \mathbb{1}(y_i > t)\frac{(1 - \hat{S}(t|\boldsymbol{x}_i))^2}{\hat{S}_C(t)}],
\end{aligned}
$$

where $\hat{S}_C(\cdot)$ is the estimated survival distribution of the censoring random variable $C$. We evaluate the Brier score at the 75th quantile of event time on the test dataset.

**Time-dependent C-index** (Antolini et al., 2005) considers a more limited yet practical set of comparable pairs compared to the conventional $C$-index, where selected subjects who developed the event earlier can't survive longer than the event horizon $t$. Formally, it is defined as

$$
C^{td}(t) = P(\hat{S}(t|\boldsymbol{x}_i) \le \hat{S}(t|\boldsymbol{x}_j)|y_i \le y_j, \delta_i = 1, y_i \le t).
$$

Following conventions, we set the event horizon at the 75th quantile of the event time, and we compute it using IPCW and truncations, aiming to obtain an unbiased estimate of $u_i < u_j$ by giving more weight to test samples with similar features that are not censored.

### 5.2. Inference Optimality on Simulated Datasets

Table 1: Summary table for simulated datasets (SD1-SD6). Sample size for each dataset is $10,000$. Event/Censored time refers to sample statistics of $Y$. The generated samples of $U$ is independently sampled across each datasets. The starting point of Gibbs sampling is fixed at $\boldsymbol{z} = (0, 0)$.

| Summary | SD1 | SD2 | SD3 | SD4 | SD5 | SD6 |
|---|---|---|---|---|---|---|
| Censor rate | 0% | 5% | 20% | 30% | 50% | 100% |
| Population mean $\mu_C$ | – | 16.00 | 8.50 | 5.50 | 0.00 | 16.00 |
| Censored time mean | – | 2.99 | -0.11 | -1.51 | -4.49 | 16.18 |
| Event time median | 1.43 | 1.29 | 0.41 | -0.30 | -3.13 | – |
| Event time min | -12.64 | -15.85 | -22.47 | -22.59 | -24.28 | – |
| Event time max | 21.27 | 21.60 | 17.49 | 18.45 | 14.29 | – |

Firstly, we investigate whether amortized CDVI can practically reduce the inference gaps compared to the vanilla VI. Table 1 provides details of 6 simulated datasets. In the simulation process, we use Gibbs sampling, where the true posterior is known and predefined. Both $P(Z|X, Y, I = 1)$ and $P(Z|X, Y, I = 0)$ are set to normal distributions, which enable the closed-form computation of the inference gaps. We vary the mean of censoring time $\mu_C$ to generate datasets with different censoring rates, in which $C$ follows an independent normal distribution. The values of the censoring rate are rounded, with an error of 1%.

Table 2: Variational inferences on simulated datasets. E-KL/C-KL: the average KL divergence between the encoder and true posterior of all events/censoring observations in the dataset. Lower is better. We set $m = k = 10$.

| Data | CD-CVAE | | CVAE | | CD-CVAE$^{+IS}$ | | CD-CVAE$^{+DVI}$ | |
|---|---|---|---|---|---|---|---|---|
| | E-KL | C-KL | E-KL | C-KL | E-KL | C-KL | E-KL | C-KL |
| SD1 | 1.65 | – | 1.65 | – | **1.53** | – | 1.56 | – |
| SD2 | **1.66** | **1.93** | 1.75 | 2.70 | 1.64 | 2.17 | 1.66 | 2.55 |
| SD3 | 2.38 | 3.13 | 2.79 | 3.18 | 2.23 | 3.21 | **2.17** | **3.13** |
| SD4 | 2.88 | 3.89 | 3.45 | 4.04 | 2.64 | 4.09 | **2.29** | **3.60** |
| SD5 | 4.45 | 5.55 | 5.42 | 5.86 | 4.11 | 5.56 | **3.89** | **5.51** |
| SD6 | – | .0871 | – | .0871 | – | .0862 | – | **.0848** |

As shown in Table 2, our proposed CD-CVAE significantly reduces the average KL divergence between the learned posterior and the true posterior in both event and censoring subsets, resulting in a smaller inference gap, which is a weighted sum of these two metrics. Leveraging VI improvement techniques (IS, DVI), CD-CVAE further reduces the inference gap across various settings of censoring. In Fig.2, we plot the marginalized learned $q_\phi$ from CVAE and CD-CVAE against the true posterior on SD4. For completeness, time-to-event modeling performance on SD3-SD5 datasets is provided in Appendix D.1.

In extreme censoring scenarios with $Y$ manually set to $U$ or $C$, CD-CVAE performs identically to CVAE, consistent with Remark 3.2. Interestingly, all models perform considerably better in the all-event scenario compared to the all-censoring scenario, and neither IS nor DVI yields significant performance improvements. Although learning a data-independent distribution of $C$ should be simpler, such a large discrepancy between these two extreme cases may imply that the amortization effect (Cremer et al., 2018) can dominate the inference gap. This observation highlights potential opportunities for practical inference improvements of amortized CDVI.

During the experiment, we also found that CD-CVAE models could converge to various local optima with nearly the same inference gap, while having different ratios of E-KL and C-KL. This observation implies a unique trade-off in the amortization CDVI, i.e., the censor/event KL trade-off.

### 5.3. Time-to-event Modeling on Benchmark Datasets

Table 3: Summary table for benchmark clinical datasets. $\bar{y}|\delta$ refers to the average event/censored survival times after applying a log transformation.

| Dataset | Size | Censored | Dim(X) | $\bar{y}|\delta = 1$ | $\bar{y}|\delta = 0$ |
|---|---|---|---|---|---|
| **SUPPORT** | 9104 | 2904 | 14 | 6.17 | 6.97 |
| **FLCHAIN** | 6524 | 4662 | 8 | 8.20 | 8.37 |
| **NWTCO** | 4028 | 3457 | 6 | 7.73 | 7.86 |
| **METABRIC** | 1980 | 854 | 8 | 7.99 | 8.14 |
| **WHAS** | 1638 | 948 | 5 | 6.95 | 7.17 |
| **GBSG** | 1546 | 965 | 7 | 3.80 | 4.18 |
| **PBC** | 418 | 257 | 17 | 4.16 | 4.32 |

Lastly, we present a comprehensive evaluation of time-to-event modeling performance, comparing CD-CVAE with state-of-the-art models. Table 3 summarizes the real-world datasets. These models include Cox-PH (Cox, 1972), Deep-Surv (Katzman et al., 2018), Deep Survival Machine (DSM) (Nagpal et al., 2021a), Random Survival Forest (RSF) (Ishwaran et al., 2008), and Deep Cox Mixture (DCM) (Nagpal et al., 2021b). All models were implemented using the Python package by Nagpal et al. (2022), and our implementation follows its API for ease of reproducibility.

Table 4 and 5 highlight CD-CVAE's performance on best $C$, $C^{td}$, and the Brier score metrics. See Appendix D.2 for full evaluations of the variant models with more baselines (Apellániz et al., 2024) and averaged metrics.

While the variants of CD-CVAE do not exhibit substantial improvements in time-to-event modeling performance, our proposed models generally outperform most of the state-of-the-art models. RSF and DCM are notably competitive in computation efficiency and hyper-parameter tuning. Nonetheless, Cox-PH with a $l_2$ regularization performs comparably on many benchmark datasets.

Table 4: Comparisons of CD-CVAE on benchmark datasets. The best model is selected by cross-validated $C$-index. The experiments are repeated five times using the same random seeds, with a train-validation-test split ratio of $0.6, 0.2, 0.2$. The highest metrics on the test dataset is reported. Higher is better: random guessing has a value of 0.5 and 1 means all comparable pairs are perfected ranked.

| Model | SUPPORT | | FLCHAIN | | NWTCO | | METABRIC | | WHAS | | GBSG | | PBC | |
|---|---|---|---|---|---|---|---|---|---|---|---|---|---|---|
| | $C$ | $C^{td}$ | $C$ | $C^{td}$ | $C$ | $C^{td}$ | $C$ | $C^{td}$ | $C$ | $C^{td}$ | $C$ | $C^{td}$ | $C$ | $C^{td}$ |
| **CoxPH** | 0.666 | 0.668 | 0.789 | 0.789 | 0.689 | 0.703 | 0.641 | 0.644 | 0.781 | 0.782 | 0.682 | 0.689 | 0.848 | 0.848 |
| **DeepSurv** | 0.648 | 0.649 | 0.780 | **0.805** | 0.674 | 0.741 | 0.664 | 0.676 | 0.786 | 0.762 | 0.609 | 0.618 | 0.855 | 0.852 |
| **DSM** | 0.666 | 0.674 | 0.801 | 0.802 | 0.706 | 0.694 | 0.666 | 0.669 | 0.811 | 0.805 | 0.615 | 0.663 | 0.862 | **0.869** |
| **RSF** | 0.683 | 0.655 | 0.768 | 0.793 | 0.677 | 0.726 | 0.686 | 0.684 | 0.808 | 0.811 | **0.706** | **0.731** | 0.857 | 0.867 |
| **DCM** | 0.682 | 0.676 | 0.788 | 0.803 | 0.680 | 0.736 | **0.689** | **0.691** | 0.803 | 0.811 | 0.625 | 0.637 | **0.866** | 0.865 |
| **CD-CVAE** | **0.685** | **0.678** | **0.811** | 0.804 | **0.708** | **0.751** | 0.681 | 0.675 | **0.868** | **0.812** | **0.706** | 0.702 | 0.863 | 0.865 |

Table 5: Comparisons of CD-CVAE in Brier Scores. The best model is selected based on the cross-validated Brier score. Experiments are repeated five times with the same random seeds, reporting the lowest test metric. Lower is better.

| Model | SUPPORT | FLCHAIN | NWTCO | MTBC | WHAS | GBSG | PBC |
|---|---|---|---|---|---|---|---|
| **CoxPH** | 0.216 | 0.121 | 0.097 | 0.214 | 0.174 | 0.222 | 0.125 |
| **DeepSurv** | **0.212** | 0.115 | 0.078 | 0.230 | 0.198 | 0.242 | 0.131 |
| **DSM** | 0.235 | 0.113 | 0.078 | 0.223 | 0.175 | 0.242 | 0.128 |
| **RSF** | 0.224 | 0.120 | 0.077 | 0.218 | **0.162** | **0.217** | **0.119** |
| **DCM** | 0.217 | 0.113 | **0.075** | 0.216 | 0.171 | 0.229 | 0.136 |
| **CD-CVAE** | 0.218 | **0.110** | **0.075** | **0.203** | 0.168 | 0.218 | 0.124 |

## 6. Related Work

**Deep Learning in Survival analysis.** Machine learning and deep learning techniques for survival analysis are not limited to LVSM. Faraggi & Simon (1995) introduced the first neural-network-based Cox regression model, allowing nonlinear relationships between covariates. A modern yet similar one is DeepSurv (Katzman et al., 2018). Deep Cox Mixture (Nagpal et al., 2021b) extends this idea to finite mixture models, but all these Coxian models rely on the proportional hazards (PH) assumption, which results in separated survival functions (Antolini et al., 2005) and may be unrealistic. A famous nonparametric tree ensemble approach, Random Survival Forest (Ishwaran et al., 2008), builds multiple decision trees to model the cumulative hazard function, leveraging the Nelson-Aalen estimator (Aalen, 1978). That said, hazard function estimation for discrete time-to-event can also be framed as a series of binary classification problems, which can be solved by black-box methods via various network architectures. DeepHit (Lee et al., 2018) uses a simple shared network to model competing risks, while RNN- (Giunchiglia et al., 2018) and Transformer-based (Hu et al., 2021) structures capture sequential relationships in time-specific predictions. These methods often require additional techniques to mitigate overfitting.

**Inference Optimality without Censoring.** Improving VI in latent variable models has been extensively studied for both labeled and unlabeled datasets. We summarize existing strategies for improving the estimate $\theta^*$ into three broad categories: (1) expanding the set of $\theta$ values for which optimal VI is attainable, i.e., increasing the support of $\theta$ where $\min_\phi B(\theta, \phi) = 0$, for example by using more expressive variational families (Ranganath et al., 2014; Kingma et al., 2016; Cremer et al., 2018); (2) modifying the variational bound to reduce $\min_\phi B(\theta, \phi)$ for general $\theta$, for example by going beyond KL-divergence and using the $\chi^2$-, $\alpha$-, or more generally, $f$-divergence (Dieng et al., 2017; Li & Turner, 2016; Wan et al., 2020). (3) adopting training techniques to improve latent disentanglement (Higgins et al., 2017), improve numeric stability (Rybkin et al., 2021), to avoid posterior collapse (Fu et al., 2019), or to better incorporate label information (Joy et al., 2021). These works require nontrivial adaptation for censoring in survival analysis. As contributions, our criticism on vanilla VI, the extensions of IS and DVI on CDVI, and the discussion on decoder variance fall under each type, respectively.

**Variational Inference for Other Tasks.** Variational methods in survival analysis are not limited to time-to-event modeling. One unsupervised task is identifying potential sub-populations, providing valuable insights for treatment recommendations and clinical decision-making (Chapfuwa et al., 2020; Franco et al., 2021; Manduchi et al., 2022; Cui et al., 2024; Jiang et al., 2024). These clustering models, if used as an intermediate step of time-to-event modeling, can be seen as a restricted LVSM, often in a D-separation latent structure. While a restrictive approach can help prevent overfitting, our criticism remains valid: the objective of VI in unsupervised tasks can be misaligned with M-estimation of the time-to-event distribution, undermining the performance of survival time prediction.

## 7. Conclusion

To the best of our knowledge, this paper has represented the first comprehensive study of variational methods for latent variable survival models. It delivers an in-depth analysis of variational inference optimality and offers valuable practical insights. The superiority of our proposed models validates a pioneering paradigm.

## Impact Statement

This paper presents work whose goal is to advance the field of Machine Learning. There are many potential societal consequences of our work, none of which we feel must be specifically highlighted here.

## Acknowledgments

This work was supported in part by NSF Grant No. SES-2316428, awarded to Xiao Wang.

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

# A. Facts and Formal Theorems

## A.1. Facts of Probability Theory

Why do we claim "CD-CVAE is interpretable as an infinite LogNormal or Weibull mixture survival regression on positive survival time"? But the choice of $\varepsilon$ that determines the location-scale family of $p_\theta(u|x,z)$ is implemented as a Normal or Gumbel-minimum distribution.

Answer: A Weibull (Lognormal) AFT of positive valued survival time $T$ is a log-linear regression assuming a Gumbel-min (Gaussian) noise (Miller, 1976). In our setting, continuous time-to-event $U$ is considered to be real-valued after the log-transform of $T$.

Table 6: Connection between AFT and the degraded LVSM

| choice of $\varepsilon$ | standarization | $p_\theta(y|\boldsymbol{x}, \boldsymbol{z})$ | linear degradation | degraded model w.r.t $T$ |
|---|---|---|---|---|
| standard Gaussian | $\tilde{y} = \frac{y - \mu_\theta(\boldsymbol{x},\boldsymbol{z})}{\sigma}$ | $\exp(-\frac{1}{2}\tilde{y}^2)/(\sqrt{2\pi}\sigma)$ | $\mu_\theta(\boldsymbol{x}, \boldsymbol{z}) = \theta^\top \boldsymbol{x}$ | Log-normal AFT |
| standard Gumbel minimum | | $\exp\{\tilde{y} - \exp(\tilde{y})\}/\sigma$ | | Weibull AFT |

## A.2. Formal Theorem 3.2.2

We have the following *Notations:*

1) The product of sets $\Phi_P(\theta) = \{(\phi_1, \phi_2) \mid \phi_1 \in \Phi_1(\theta), \phi_2 \in \Phi_2(\theta)\}$ denotes the set of optimal parameters of $q_{\phi_1,\phi_2}$, and $\Theta_P = \{\theta \mid \Phi_P(\theta) \neq \varnothing\}$ denotes its support.

2) $\Phi_{EU}(\theta) = \{(\phi, \phi)|\phi \in \Phi_1(\theta, \sigma) \cap \Phi_2(\theta, \sigma)\}$ denotes the embedding set of $\Phi_U(\theta)$. The support $\Theta_{EU} = \{\theta \mid \Phi_{EU}(\theta) \neq \varnothing\}$. For any $\theta$, $\Phi_{EU}(\theta) \subseteq \Phi_P(\theta)$. Optimal $q_{\phi_1,\phi_2}$ is degenerated to the optimal $q_\phi$ in Vanilla VI if $(\phi_1, \phi_2) \in \Phi_{EU}(\theta)$.

**Theorem 3.2.2 (Inference optimality of CDVI).**
*Following the assumptions. If $\Theta_P \neq \varnothing$, then*

(5) **Constraint on optimal $\phi_1$ and $\phi_2$.** $\forall(\phi_1, \phi_2) \in \Phi_P(\theta), q_{\phi_1}(z|x,y) \propto_z h_\theta(y|x,z)q_{\phi_2}(z|x,y)$.

*If $\Theta_P \backslash \Theta_{EU} \neq \varnothing$, we have the following results.*

(6) **Strict better optimal $\phi$.** $\forall\theta \in \Theta_P \backslash \Theta_{EU}$, we have $\varnothing = \Phi_{EU}(\theta) \subset \Phi_P(\theta)$, and more importantly, $\forall(\phi_1, \phi_2), s.t.\ \phi_1 = \phi_2, \exists(\phi_1^*, \phi_2^*) \in \Phi_P(\theta)$,

$$L(\theta) = \text{ELBO-C}(\theta, \phi_1^*, \phi_2^*) > \text{ELBO}(\theta, \phi_1, \phi_2).$$

(7) **Non-degraded location parameter.** *If $U|X, Z$ is a location-scale distribution parameterized by the location parameter $\mu_\theta(x, z)$ and the deterministic scale parameter $\sigma$, then for all $\theta \in \Theta_P \backslash \Theta_{EU}$, there exists $z_1 \neq z_2$, $\mu_\theta(x, z_1) \neq \mu_\theta(x, z_2)$ for almost all $x$*

(8) **Lazy posterior free.** $\forall\theta \in \Theta_P \backslash \Theta_{EU}, \forall(\phi_1, \phi_2) \in \Phi_P(\theta)$, *such that*

$$\delta\text{KL}[q_{\phi_1}\| p_\theta(z|x)] + (1 - \delta)\text{KL}[q_{\phi_2}\| p_\theta(z|x)] > 0.$$

*If $z \perp\!\!\!\perp x$ is assumed, i.e., $p_\theta(z|x) = p(z)$, optimal censor-dependent VI is posterior collapse free.*

Theorem 3.2.2 demonstrates how the CDVI resolves the issues of vanilla VI. Claim (5) states that the optimal $q_{\phi_1,\phi_2}$ captures the constraints on the parameters $\phi_1$ and $\phi_2$, preventing it from being reduced to the naive $q_\phi$. As we show in Remark 3.2, the assumption of $\phi_1 = \phi_2$ is the root cause of latent non-identifiability in Proposition 3.1. Claim (6) shows that the optimal $q_{\phi_1,\phi_2}$ enjoys expanded support $\Theta_P$, enabling our $q_{\phi_1,\phi_2}$ to achieve VI optimality at specific $\theta$ values where vanilla VI would fail. To be specific, Claim (7) demonstrates that $\theta$ maintains the complexity and expressive power of the latent variable model $f_\theta(y|x)$. Consequently, Claim (8) shows that the optimal $q_{\phi_1}$ or $q_{\phi_2}$ will not remain lazy or suffer from the posterior collapse issue.

## A.3. Formal Theorem 4.3.3

Following the definition 4.3.2, let $\alpha_i := \mathbb{E}[(\hat{f}_m - f)^i]$, and $\beta_i := \mathbb{E}[(\hat{S}_k - S)^i]$ be the $i^{th}$ central moments of unbiased estimators $\hat{f}_m$ and $\hat{S}_k$. Obviously, $\alpha_1 = 0, \beta_1 = 0$.

**Lemma 1 (Asymptotic bias of** $\log \hat{f}_m$**, and** $\log \hat{S}_k$**).** *If $f$ and $\alpha_i$ is finite for all $i \geq 1$, then for $m \to \infty$,*

$$\mathbb{E}_{\boldsymbol{z}}[\log \hat{f}_m] = \log f - \frac{1}{m}\frac{\alpha_2}{2f^2} + \frac{1}{m^2}(\frac{\alpha_3}{3f^3} - \frac{3\alpha_2}{4f^4}) + o(m^{-2}). \tag{19}$$

*If $S$ and $\beta_i$ are finite for all $i \geq 1$, then for $k \to \infty$,*

$$\mathbb{E}_{\boldsymbol{z}}[\log \hat{S}_k] = \log S - \frac{1}{k}\frac{\beta_2}{2S^2} + \frac{1}{k^2}(\frac{\beta_3}{3S^3} - \frac{3\beta_2}{4S^4}) + o(k^{-2}). \tag{20}$$

*The expectation is taken over $\boldsymbol{z}_{1:m}$, or $\boldsymbol{z}_{1:k}$ for any given $\boldsymbol{x}, y, \theta, \phi_1, \phi_2$.*

Lemma 1 demonstrates the asymptotic bias of importance sampling induced loglikelihood estimators $\log \hat{f}_m$ and $\log \hat{S}_k$, which has an order of magnitude of $m^{-1}$ or $k^{-1}$.

**Lemma 2 (Asymptotic variance of** $\log \hat{f}_m$**, and** $\log \hat{S}_k$**).** *If $f$ and $\alpha_i$ are finite for all $i \geq 1$, then for $m \to \infty$,*

$$\mathbb{V}[\log \hat{f}_m] = \frac{1}{m}\frac{\alpha_2}{f^2} - \frac{1}{m^2}(\frac{\alpha_3}{f^3} - \frac{5\alpha_2}{f^4}) + o(m^{-2}). \tag{21}$$

*Similarly, if $S$ and $\beta_i$ are finite for all $i \geq 1$, then for $k \to \infty$,*

$$\mathbb{V}[\log \hat{S}_k] = \frac{1}{k}\frac{\beta_2}{S^2} - \frac{1}{k^2}(\frac{\beta_3}{S^3} - \frac{5\beta_2}{S^4}) + o(k^{-2}). \tag{22}$$

Recall that $\hat{L}_{m,k} := \delta \log \hat{f}_m + (1 - \delta) \log \hat{S}_k$, we use lemma 1 & 2 to get the following.

**Lemma 3 (Asymptotic bias of** $\hat{L}_{m,k}$**).** *Under the assumption of Lemma 1, for $m, k \to \infty$,*

$$\begin{aligned}
L(\theta) - \text{ELBO-C}_{m,k} = {}& \delta[\frac{1}{m}\frac{\alpha_2}{2f^2} - \frac{1}{m^2}(\frac{\alpha_3}{3f^3} - \frac{3\alpha_2}{4f^4})] \\
& + (1 - \delta)[\frac{1}{k}\frac{\beta_2}{2S^2} - \frac{1}{k^2}(\frac{\beta_3}{3S^3} - \frac{3\beta_2}{4S^4})] + o(m^{-2}) + o(k^{-2}).
\end{aligned} \tag{23}$$

**Lemma 4 (Asymptotic variance of** $\hat{L}_{m,k}$**).** *Under the assumption of Lemma 1, for $m, k \to \infty$,*

$$\mathbb{E}_z[(\hat{L}_{m,k} - L(\theta))^2] = \delta[\frac{1}{m}\frac{\alpha_2}{f^2} - \frac{1}{m^2}(\frac{\alpha_3}{f^3} - \frac{20\alpha_2 + \alpha_2^2}{4f^4})] + (1 - \delta)[\frac{1}{k}\frac{\beta_2}{S^2} - \frac{1}{k^2}(\frac{\beta_3}{S^3} - \frac{20\beta_2 + \beta_2^2}{4S^4})] + o(m^{-2}) + o(k^{-2}). \tag{24}$$

**Theorem 4.3.3 (Formal; Consistency of** $\hat{L}_{m,k}$**).**
*Under the assumption in Lemma 1, for $m \to \infty, k \to \infty$, for all $\xi > 0$,*

$$\lim_{m,k \to \infty} P(|\hat{L}_{m,k} - L(\theta)| > \xi) = 0.$$

The proof is almost a direct result of Lemma 3 and Lemma 4.

## A.4. Theorem for Delta method CDVI

We prove that the Delta method CDVI yields a smaller asymptotic inference gap/bias, as we mentioned after Definition 4.3.2. Following Eq.17 and Eq.18, let $\dot{L}_{m,k} := \delta \log \dot{f}_m + (1 - \delta) \log \dot{S}_k$.

**Theorem A.4 (less asymptotic bias of delta method CDVI).**
*Under the assumption in Lemma 1, for $m \to \infty, k \to \infty$,*

$$\mathbb{E}[\frac{\hat{\alpha_2}}{2m(\hat{f}_m)}] = \frac{\alpha_2}{2mf^2} - \frac{1}{m^2}(\frac{\alpha_3}{f^3} - \frac{3\alpha_2^2}{2f^4}) + o(m^{-2}), \quad \mathbb{E}[\frac{\hat{\beta_2}}{2k(\hat{S}_k)}] = \frac{\beta_2}{2kS^2} - \frac{1}{k^2}(\frac{\beta_3}{S^3} - \frac{3\beta_2^2}{2S^4}) + o(k^{-2}),$$

*and*

$$L(\theta) - \mathbb{E}_{\boldsymbol{z}}[\log \dot{L}_{m,k}] = \delta[\frac{1}{m^2}(\frac{2\alpha_3}{3f^3} - \frac{3\alpha_2}{4f^4})] + (1 - \delta)[\frac{1}{k^2}(\frac{2\beta_3}{3S^3} - \frac{3\beta_2}{4S^4})] + o(m^{-2}) + o(k^{-2}).$$

Compared with Lemma 3, the asymptotic inference gap/bias is reduced by one order of magnitude of $m$ and $k$.

# B. Proofs

## B.1. Proof for Proposition 3.1

Proof of (1): From Lemma 3.1 and assumption 1), for any optimal parameter $\phi \in \Phi_U$, we have

$$q_\phi(\boldsymbol{z}|\boldsymbol{x}, u) = f_\theta(u|\boldsymbol{x}, \boldsymbol{z})p_\theta(\boldsymbol{z}|\boldsymbol{x})/f_\theta(u|\boldsymbol{x}) = S_\theta(u|\boldsymbol{x}, \boldsymbol{z})p_\theta(\boldsymbol{z}|\boldsymbol{x})/S_\theta(u|\boldsymbol{x}),$$

which means for any $\boldsymbol{z}$,

$$h_\theta(u|\boldsymbol{x}, \boldsymbol{z}) = f_\theta(u|\boldsymbol{x}, \boldsymbol{z})/S_\theta(u|\boldsymbol{x}, \boldsymbol{z}) = f_\theta(u|\boldsymbol{x})/S_\theta(u|\boldsymbol{x}) = h_\theta(u|\boldsymbol{x}).$$

Proof of (2): Since $U$ is continuous, $h()$ in the above equation can be replaced by $f, F, S, H$ due to the 1-1 relationship, e.g. $h(u) = -\frac{\partial \log S(u)}{u}$, leading to $f_\theta(u|x, z) = f_\theta(u|x)$.

Since $f_\theta(u|x, z) \stackrel{d}{=} \mu_\theta(x, z) + \sigma \times \varepsilon$, we claim that $\mu(x, z)$ is independent of the value of $z$

Proof of (3): Also based on $f_\theta(u|x, z) = f_\theta(u|x)$ for any $z$,

$$q_\phi(\boldsymbol{z}|\boldsymbol{x}, u) = f_\theta(u|\boldsymbol{x}, \boldsymbol{z})p_\theta(\boldsymbol{z}|\boldsymbol{x})/f_\theta(u|\boldsymbol{x}) = p_\theta(\boldsymbol{z}|\boldsymbol{x}).$$

Proof of (4): If a V-structure latent graph is assumed, i.e., $\boldsymbol{z} \perp\!\!\!\perp \boldsymbol{x}$, then the prior $p_\theta(z|x) = p(z)$.

Now, (4) can be drawn immediately from the conclusion of (3).

## B.2. Proof for Theorem 3.2.1

In the section 2.1, we mentioned that the partial likelihood $f_\theta(y|\boldsymbol{x})^\delta S_\theta(y|\boldsymbol{x})^{1-\delta}$, although it contains all the information of $\theta$, is not a proper density. Here we further emphasize that the appropriate variational distribution cannot be discussed separately on the subspace $\mathcal{D}_E$ and $\mathcal{D}_C$ targeting distribution function $f_\theta(y|\boldsymbol{x})$ and $S_\theta(y|\boldsymbol{x})$ in the vanilla VI framework (Xiu et al., 2020; Nagpal et al., 2021a), because it leads to ignoring the information of $\delta$.

Proof: Abusing the "density" notation $p(y, \delta, z|x)$ and $p(y, \delta|x)$ for the Radon-Nikodym derivative of $P(Y, I, Z|X)$ and $P(Y, I|X)$, the general variational bound defined in Domke & Sheldon (2018) is

$$\log p(y, \delta|x) = \underbrace{\mathbb{E}_z[\log R]}_{\text{bound}} + \underbrace{\mathbb{E}_z[\log \frac{p(y, \delta|x)}{R}]}_{\text{looseness}}.$$

For a simple non-augmented variational bound enabling Jensen's inequality, $R$ should be

$$R(z) = \frac{p(y, \delta, z|x)}{q_\phi(z)}.$$

A tight "looseness" requires the KL divergence being zero, leading to optimal $q_{\phi^*}(z) := p(z|x, y, \delta)$, which is the density of $P(Z|X, Y, I)$ parameterized by both $\theta$ from $U$ and $\eta$ from $C$.

Now we prove that both $p(z|x, y, \delta = 1)$ and $p(z|y, \delta = 0, x)$ will be independent of $C$ and free from $\eta$. Assuming that 1) continuous $U|X, C|X$ have the same support $\mathcal{U}$, 2) conditional independent censoring, 3) independence between $C$ and $Z$ given $X$, 4) Fubini's theorem is applicable, we have

$$p(z|x, y, \delta = 1) = \frac{p(y, \delta = 1, z|x)}{p(y, \delta = 1|x)} = \frac{p_{U,Z}(y, z|\boldsymbol{x})P(C \geq y|x)\mathbb{1}(y \in \mathcal{U})}{p_U(y|\boldsymbol{x})P(C \geq y|x)\mathbb{1}(y \in \mathcal{U})} = \frac{p_U(y|\boldsymbol{x}, \boldsymbol{z})p(\boldsymbol{z}|\boldsymbol{x})}{p_U(y|\boldsymbol{x})}\mathbb{1}(y \in \mathcal{U}).$$

Reorganizing terms, we get $q_{\phi^*}(z|x, y, \delta = 1) = p(z|x, y, \delta = 1) = f_\theta(y, z|x)/f_\theta(y|x)$. We use the similar proof for $q_\phi(z|x, y, \delta = 0)$. We note that the assumption of the same support is inadmissible, and we can also express $q_\phi^*$ as follows

$$q_{\phi^*}(z|x, y, \delta) = q_{\phi^*}(z|x, y, \delta = 1)^\delta q_{\phi^*}(z|x, y, \delta = 0)^{1-\delta},$$

which leads to the notation of Definition 3.2.

Proof for Remark 3.2 follows naturally. The marginalized $q_{\phi^*}(z|x, y) = p(z|x, y, \delta = 1) * P(\delta = 1|x, y) + p(z|x, y, \delta = 0) * P(\delta = 0|x, y)$, which will not equal $p(z|x, y, \delta = 1)$ or $p(z|x, y, \delta = 0)$ unless one of $P(\delta|x, y)$ is zero.

### B.3. Proof for Formal Theorem 3.2.2

Proof of (5): Using the above conclusion of $q_{\phi^*}$,

$$h_\theta(y|x,z) := \frac{f_\theta(y|x,z)}{S_\theta(y|x,z)} = \frac{q_{\phi^*}(z|x,y,\delta=1) * f_\theta(y|x)/p(z|x)}{q_{\phi^*}(z|x,y,\delta=0) * S_\theta(y|x)/p(z|x)}.$$

Denoting $q_{\phi^*}(z|x,y,\delta=1)$ by $q_{\phi_1^*}(z|x,y)$ and reorganizing the terms, we have

$$q_{\phi_1^*}(z|x,y) = \frac{1}{h_\theta(y|x)} q_{\phi_2^*}(z|x,y) h_\theta(y|x,z) \propto_z q_{\phi_2^*}(z|x,y) h_\theta(y|x,z).$$

Proof of (6): The proof is trivial, following the definition of optimal $q_{\phi^*}(z|x,y,\delta)$.

Proof of (7): We have already proved in (1) that if $\theta \in \Theta_p$ then $f_\theta(u|x,z) = f_\theta(u|x)$ and also in (2) that if $f_\theta(u|x,z)$ is a location-scale family, it leads to $\mu_\theta(x,z)$ independent of $z$. We now prove the reverse is also true: if $\mu_\theta(x,z)$ is independent of $z$, and $f(u|x,z)$ the density of location-scale family, it leads to $f_\theta(u|x,z) = f_\theta(u|x)$; $S_\theta(u|x,z) = S_\theta(u|x)$, thus we have

$$q_{\phi^*}(z|x,y,\delta=0) = q_{\phi^*}(z|x,y,\delta=1) = p_\theta(z|x).$$

Equivalently, following the notation, we have $\phi_1^* = \phi_2^*$. Thus, $\phi_1^*, \phi_2^* \in \Phi_{EU}$, meaning that $\theta \in \Theta_{EU}$.

Then we complete the proof by contrapositive.

Proof of (8): By non-negativity of KL divergence, the KL divergence is zero if and only if the above equation in (7) holds true. Thus, it is a direct result of (7). If V-structure is assumed, prior $p_\theta(z|x)$ is replaced by $p(z)$.

### B.4. Proof for Proposition 4.2

Here, we prove that if $\varepsilon$ is standard normal or standard Gumbel-minimum distribution, there is no closed-form solution of $\partial\text{ELBO-C}/\partial\sigma$ given the parameter of $\zeta, \phi_1, \phi_2$. For notation clarity, we decompose $\theta = (\zeta, \sigma)$ where $\mu_\zeta(x,z)$ is the location parameter of the decoder, and $\sigma$ is its scale parameter.

Proof: Notice that KL divergence terms in ELBO-C do not involve $\sigma$, and the expectation is taken over $q_{\phi_1,\phi_2}$. Given dataset $\{x_i, y_i, \delta_i\}_{i=1}^n$

$$\frac{\partial\text{ELBO-C}}{\partial\sigma} = \mathbb{E}\Big[\sum_{i:\delta_i=1} \frac{\partial \log f_\theta(y_i|x_i,z)}{\partial\sigma} + \sum_{i:\delta_i=0} \frac{\partial \log S_\theta(y_i|x_i,z)}{\partial\sigma}\Big].$$

Using chain rule and the density in Table 6, we have the following result:

(1) If the decoder is normal, these two terms can be expressed as

$$\frac{\partial \log f_\theta(y_i|x_i,z)}{\partial\sigma} = -\frac{1}{\sigma} + \frac{\tilde{y}_i^2}{\sigma}, \quad \frac{\partial \log S_\theta(y_i|x_i,z)}{\partial\sigma} = \frac{\partial \log 1 - \Phi(\tilde{y}_i)}{\partial\tilde{y}_i} * \frac{\partial\tilde{y}_i}{\partial\sigma} = \lambda(\tilde{y}_i) * \frac{\tilde{y}_i}{\sigma},$$

where $\lambda(s)$ is the hazard function of the standard normal distribution that 1) has no closed-form expression, 2) is convex, 3) can be bounded. One naive bound is $\lambda(s) > s$; a tighter bound $\lambda(s) \geq \frac{3}{4s} + \frac{\sqrt{s^2+8}}{4}$ for $s > 0$ is provided by Baricz (2008) via Mill's ratio Mitrinovic (1970)

(2) If the decoder is Gumbel-minimum ($S(s) = \exp(-\exp(s))$), these two terms can be expressed as

$$\frac{\partial \log f_\theta(y_i|x_i,z)}{\partial\sigma} = \frac{\partial(\tilde{y}_i - \exp(\tilde{y}_i))}{\partial\tilde{y}_i}\frac{\partial\tilde{y}_i}{\partial\sigma} - \frac{1}{\sigma} = -\frac{\tilde{y}_i+1}{\sigma} + \exp(\tilde{y}_i)\frac{\tilde{y}_i}{\sigma}, \quad \frac{\partial \log S_\theta(y_i|x_i,z)}{\partial\sigma} = \frac{\partial - \exp(\tilde{y}_i)}{\partial\tilde{y}_i} * \frac{\partial\tilde{y}_i}{\partial\sigma} = \exp(\tilde{y}_i)\frac{\tilde{y}_i}{\sigma}.$$

Neither of these expressions leads to a closed form solution of $\sigma$ when $\frac{\partial\text{ELBO-C}}{\partial\sigma} = 0$.

## B.5. Proof for Theorem ??

The proof mainly follows (Burda et al., 2015): we are going to prove the monotonicity of ELBO-C$_{m,k}$, instead of $B(m,k)$.

Proof: Given $m, k$, let $m', k'$ be any integers less than $m, k$, respectively. Denote the subset of index $I_{m'} = \{i_1, ..., i_{m'}\} \subset \{1, 2, 3, .., m\}$ as a uniformly distributed subset of distinct indices where $|I_{m'}| = m'$. $I_{k'}$ follows the same definition for $\{1, 2, 3, .., k\}$. For any bounded sequence of $a_1, ..., a_m$,

$$\mathbb{E}_{I_{m'}}\left[\frac{a_{i_1} + ... + a_{i_{m'}}}{m'}\right] = \frac{a_1 + a_2 + ... + a_m}{m}.$$

Therefore,

$$\begin{aligned}
\text{ELBO-C}_{m,k} &:= \mathbb{E}[\log(\hat{f}_m^\delta \hat{S}_k^{1-\delta})] = \delta\mathbb{E}_{z_{1:m}}[\log(\hat{f}_m)] + (1-\delta)\mathbb{E}_{z_{1:k}}[\log(\hat{S}_k)] \\
&= \delta\mathbb{E}_{z_{1:m}}\left[\log\frac{\hat{f}_1(z_1) + \hat{f}_1(z_2) + .. + \hat{f}_1(z_m)}{m}\right] + (1-\delta)\mathbb{E}_{z_{1:k}}[\log(\hat{S}_k)] \\
&= \delta\mathbb{E}_{z_{1:m}}\left[\log\mathbb{E}_{I_{m'}}\frac{\hat{f}_1(z_{i_1}) + \hat{f}_1(z_{i_2}) + .. + \hat{f}_1(z_{i_{m'}})}{m'}\right] + (1-\delta)\mathbb{E}_{z_{1:k}}[\log(\hat{S}_k)] \\
&\geq \delta\mathbb{E}_{z_{1:m}}\left[\mathbb{E}_{I_{m'}}\left[\log\frac{\hat{f}_1(z_{i_1}) + \hat{f}_1(z_{i_2}) + .. + \hat{f}_1(z_{i_{m'}})}{m'}\right]\right] + (1-\delta)\mathbb{E}_{z_{1:k}}[\log(\hat{S}_k)] \\
&= \delta\mathbb{E}_{z_{1:m'}}[\log\hat{f}_{m'}]] + (1-\delta)\mathbb{E}_{z_{1:k}}[\log(\hat{S}_k)] = \text{ELBO-C}_{m',k}.
\end{aligned}$$

Similarly, ELBO-C$_{m,k} \geq$ ELBO-C$_{m,k'}$. Thus,

$$\text{ELBO-C}_{m,k} \geq \max(\text{ELBO-C}_{m',k}, \text{ELBO-C}_{m,k'}) > \min(\text{ELBO-C}_{m',k}, \text{ELBO-C}_{m,k'}) \geq \text{ELBO-C}_{m',k'} \geq \text{ELBO-C}.$$

Here ELBO-C$_{m,k} \leq L(\theta)$ is ensured by Jensen's inequality and Fubini's theorem.

Assuming a bounded $\hat{f}_1; \hat{S}_1$, we use the strong law of large numbers: for $m \to \infty, \hat{f}_m \overset{a.s.}{\to} \mathbb{E}[\hat{f}_1] = f$.

Similar results apply to $\hat{S}_k$. The results imply convergence in expectation

$$\lim_{m\to\infty, k\to\infty} \text{ELBO-C}_{m,k} = L(\theta).$$

Then, using the definition of $B(m,k) := L(\theta) - \text{ELBO-C}_{m,k}$, we complete the proof by reversing the inequality.

## B.6. Proof for Theorem 4.3.2

This theorem is a **corrected** extension of Theorem 1 in Domke & Sheldon (2018). Our proof follows a similar structure, but we first highlight the mistake in the original proof in Theorem 1 in Domke & Sheldon (2018). In their original proof, the definition of Eq.5 is not consistent in Theorem 1, where the expectation in Eq.5 is taken over $z_1, ..z_m$ from $q_\phi$, while the expectation in Theorem 1 is taken over the augmented variational distribution, as shown on Page 15. Thus, when generalizing their results, we must warn the reader that the KL divergence term does not correspond to the inference gap defined in Thm. 4.3.1

Proof: Some basic facts from the definition: 1) $Q_1(1) = q_{\phi_1}(z|x,y), Q_2(1) = q_{\phi_2}(z|x,y)$; 2) $P_1(1) = f_\theta(y,z|x)$; 3) $P_2(1) = S_\theta(y|x,z)p_\theta(z|x)$; 4) For any $m > 0, \int\int J_1(m)dz_{1:m} = \log f(y|x)$; 5) For any $k > 0, \int J_2(k)dz_{1:k} = \log S(y|x)$.

Since $\log p(x) = E_{q(z)}\log\frac{p(x,z)}{q(z)} + \text{KL}[q(z)||p(z|x)]$, we have

$$\log f(y|x) = \mathbb{E}_{Q_1(m)}\log\frac{J_1(m)}{Q_1(m)} + \text{KL}[Q_1(m)||P_1(m)]; \quad \log S(y|x) = \mathbb{E}_{Q_2(m)}\log\frac{J_2(m)}{Q_2(m)} + \text{KL}[Q_2(m)||P_2(m)].$$

By definition, we have

$$\mathbb{E}_{Q_1(m)}\log\frac{J_1(m)}{Q_1(m)} = \mathbb{E}_{Q_1(m)}\left[\log\frac{J_1(m)}{J_1(m)/\hat{f}_m}\right] = \mathbb{E}_{Q_1(m)}[\log\hat{f}_m].$$

Similar results for $Q_2(m)$ can be obtained. Adding these two equations with multiplication of $\delta$ or $1 - \delta$, we have

$$L(\theta) := \delta \log f(y|x) + (1 - \delta) \log S(y|x) = \mathbb{E}_{Q_1, Q_2}[\log \hat{f}_m^\delta \hat{S}_k^{1-\delta}] + \text{KL}[Q_1(m)||P_1(m)]^\delta \text{KL}[Q_2(k)||P_2(k)]^{(1-\delta)}.$$

The above equation completes the proof. We highlight that the mentioned mistake limits further interpretation. It is easy to see that

$$B(1,1) := \text{KL}[Q_1(1)||P_1(1)]^\delta \text{KL}[Q_2(1)||P_2(1)]^{1-\delta}.$$

However, we cannot subtract these two KL divergences by the chain rule of KL divergence as in Domke & Sheldon (2018). Since $L(\theta) - \mathbb{E}_{Q_1, Q_2}[\log \hat{f}_m^\delta \hat{S}_k^{1-\delta}]$ does not correspond to $B(m, k)$, the subtraction does not give any meaningful interpretation.

### B.7. Proof for Lemma 1-6 and Formal Theorem 4.3.3

The chain of proofs follows the same structure as (Nowozin, 2018), with minor corrections and better consistency of notations.

#### B.7.1. PROOF OF LEMMA 1

Given the assumptions, which are sufficient for Fubini's theorem to apply, the Taylor expansion of $\mathbb{E}[\log \hat{f}]$ at $\log f$ is given as

$$\mathbb{E}[\log \hat{f}_m] = \mathbb{E}[\log(f - (\hat{f}_m - f))] = \log f - \sum_{i=1}^{\infty} \frac{(-1)^i}{i f^i} \mathbb{E}[(\hat{f}_m - f)^i] := \log f - \sum_{i=1}^{\infty} \frac{(-1)^i}{i f^i} \alpha_i'.$$

From Theorem 1 of Angelova (2012), using the definition of $\alpha_i, \beta_i$, we can get the relationship between $\alpha_i'$ and $\alpha_i$:

$$\alpha_2' = \frac{\alpha_2}{m}; \alpha_3' = \frac{\alpha_3}{m^2}; \alpha_4' = \frac{3}{m^2}\alpha_2^2 + o(m^{-2}).$$

By substituting $\alpha_i'$ with $\alpha_i$ we have

$$\mathbb{E}[\log \hat{f}_m] = \log f - \frac{1}{2f^2}\frac{\alpha_2}{m} + \frac{1}{3f^3}\frac{\alpha_3}{m^2} - \frac{1}{4f^4}\left(\frac{3}{m^2}\alpha_2^2\right) + o(m^{-2}).$$

After rearrangement, we complete the proof for $\mathbb{E}[\log \hat{f}_m]$. By applying the same proof as for $\mathbb{E}[\log \hat{S}_k]$, we complete the whole proof. We denote $\mathbb{B}[\log \hat{f}_m] = \frac{1}{2f^2}\frac{\alpha_2}{m} - \frac{1}{3f^3}\frac{\alpha_3}{m^2} + \frac{1}{4f^4}\left(\frac{3}{m^2}\alpha_2^2\right)$ and similarly for $\mathbb{B}[\log \hat{S}_k]$.

#### B.7.2. PROOF OF LEMMA 2

By the definition of variance and using the same expansion on both $\log \hat{f}_m$ and its expectation at $\log f$, we have

$$\mathbb{V}[\log \hat{f}_m] = \mathbb{E}[(\log \hat{f}_m - \mathbb{E} \log \hat{f}_m)^2] = \mathbb{E}\left[\left(\sum_{i=1}^{\infty} \frac{(-1)^i}{i f^i}(\mathbb{E}[(\hat{f}_m - f)^i] - (\hat{f}_m - f)^i)\right)^2\right].$$

By expanding the above equation to the third order, we have

$$\mathbb{V}[\log \hat{f}_m] \approx \frac{\alpha_2'}{f^2} - \frac{1}{f^3}(\alpha_3' - \alpha_1'\alpha_2') + \frac{2}{3f^4}(\alpha_4' - \alpha_1'\alpha_3') + \frac{1}{4f^4}(\alpha_4' - (\alpha_2')^2) - \frac{1}{3f^5}(\alpha_5' - \alpha_2'\alpha_3') + \frac{1}{9f^6}(\alpha_6' - (\alpha_3')^2).$$

By substituting $\alpha_i'$ with $\alpha_i$, we complete the proof for $\mathbb{V}[\log \hat{f}_m]$. By applying the same proof as for $\mathbb{V}[\log \hat{S}_k]$, we complete the whole proof.

### B.7.3. PROOF OF LEMMA 3

Notice that $\delta$ is binary valued and finite, thus for $m \to \infty$ and $k \to \infty$, where the sequence of limitation doesn't matter, we have

$$
\begin{aligned}
\text{ELBO-C}_{m,k} = \mathbb{E}[\hat{L}_{m,k}] &= \delta\mathbb{E}[\log \hat{f}_m] + (1-\delta)\mathbb{E}[\log \hat{S}_k] \\
&= \delta \log f + (1-\delta)\log S - \delta[\frac{1}{m}\frac{\alpha_2}{2f^2} - \frac{1}{m^2}(\frac{\alpha_3}{3f^3} - \frac{3\alpha_2}{4f^4})] \\
&\quad + (1-\delta)[\frac{1}{k}\frac{\beta_2}{2S^2} - \frac{1}{k^2}(\frac{\beta_3}{3S^3} - \frac{3\beta_2}{4S^4})] + o(m^{-2}) + o(k^{-2}).
\end{aligned}
$$

By substituting $L(\theta) = \delta \log f + (1-\delta)\log S$, we complete the proof.

### B.7.4. PROOF OF LEMMA 4

Notice that 1) $\phi_1, \phi_2$ here are not optimally constrained in Claim (5) of Theorem 3.2.2, 2) the expectation w.r.t $z_{1:m}$ and $z_{1:k}$ can be separated due to independence, 3) $L(\theta)$ is not a function of $z$, and 4) $\delta^2 = \delta$. For $m \to \infty$, $k \to \infty$,

$$
\begin{aligned}
\mathbb{E}[(\hat{L}_{m,k} - L(\theta))^2] &= \mathbb{E}_{z_{1:m}}\left[\left(\delta \log \hat{f}_m - \delta \log f\right)^2\right] + \mathbb{E}_{z_{1:k}}\left[\left((1-\delta)\log \hat{S}_k - (1-\delta)\log S\right)^2\right] \\
&= \delta\mathbb{E}[(\log \hat{f}_m - \mathbb{E}[\log \hat{f}_m] + \mathbb{E}[\log \hat{f}_m] - \log f)^2] + (1-\delta)\mathbb{E}[(\log \hat{S}_k - \mathbb{E}[\log \hat{S}_k] + \mathbb{E}[\log \hat{S}_k] - \log S)^2] \\
&= \delta\mathbb{V}[\log \hat{f}_m] + \delta(\mathbb{B}[\log \hat{f}_m])^2 + (1-\delta)\mathbb{V}[\log \hat{S}_k] + (1-\delta)(\mathbb{B}[\log \hat{S}_m])^2 \\
&= \delta[\frac{1}{m}\frac{\alpha_2}{f^2} - \frac{1}{m^2}(\frac{\alpha_3}{f^3} - \frac{5\alpha_2}{f^4}) + (\frac{1}{m}\frac{\alpha_2}{2f^2})^2] + (1-\delta)[\frac{1}{k}\frac{\beta_2}{S^2} - \frac{1}{k^2}(\frac{\beta_3}{S^3} - \frac{5\beta_2}{S^4}) + (\frac{1}{k}\frac{\beta_2}{2S^2})^2] + o(m^{-2}) + o(k^{-2}) \\
&= \delta[\frac{1}{m}\frac{\alpha_2}{f^2} - \frac{1}{m^2}(\frac{\alpha_3}{f^3} - \frac{20\alpha_2 + \alpha_2^2}{4f^4})] + (1-\delta)[\frac{1}{k}\frac{\beta_2}{S^2} - \frac{1}{k^2}(\frac{\beta_3}{S^3} - \frac{20\beta_2 + \beta_2^2}{4S^4})] + o(m^{-2}) + o(k^{-2}).
\end{aligned}
$$

### B.7.5. PROOF OF FORMAL THEOREM 4.3.3

Proof:

$$
\begin{aligned}
P(|\hat{L}_{m,k} - L(\theta)| \geq \xi) &= P(|\hat{L}_{m,k} - \mathbb{E}[\hat{L}_{m,k}] + \mathbb{E}[\hat{L}_{m,k}] - L(\theta)| \geq \xi) \\
&\leq P(|\hat{L}_{m,k} - \mathbb{E}[\hat{L}_{m,k}]| + |\mathbb{E}[\hat{L}_{m,k}] - L(\theta)| \geq \xi) \\
&\leq \underbrace{P(|\hat{L}_{m,k} - \mathbb{E}[\hat{L}_{m,k}]| \geq \xi/2)}_{①} + \underbrace{P(|\mathbb{E}[\hat{L}_{m,k}] - L(\theta)| \geq \xi/2)}_{②}.
\end{aligned}
$$

(25)

Notice that $|\mathbb{E}[\hat{L}_{m,k}] - L(\theta)|$ is not random, and based on the result of Lemma 3, for sufficiently large $m_1, k_1$, we have $|\mathbb{E}[\hat{L}_{m,k}] - L(\theta)| < \xi/2$, regardless of the value of $\delta$. This proves that $② \to 0$ as $m, k \to \infty$

By Chebyshev's inequalities,

$$
P(|\hat{L}_{m,k} - \mathbb{E}[\hat{L}_{m,k}]| \geq \xi/2) \leq \frac{4}{\xi^2}\mathbb{V}[\hat{L}_{m,k}].
$$

Based on the result of Lemma 4, we have $\frac{4}{\xi^2}\mathbb{V}[\hat{L}_{m,k}] \to 0$ as $m, k \to \infty$, regardless of the value of $\delta$. This proves that $① \to 0$ as $m, k \to \infty$.

Together, we establish the convergence in probability and hence consistency of $\hat{L}_{m,k}$.

### B.7.6. PROOF OF THEOREM A.4

Following Definition 4.3.2, we consider the induced log-likelihood estimator

$$
\dot{L}_{m,k} = \delta \log \dot{f}_m + (1-\delta)\log \dot{S}_k.
$$

Recall lemma 1, we have proved that

$$\mathbb{E}[\log \hat{f}_m] = \log f - \frac{1}{m}\frac{\alpha_2}{2f^2} + \frac{1}{m^2}(\frac{\alpha_3}{3f^3} - \frac{3\alpha_2^2}{4f^4}) + o(m^{-2}).$$

By Definition 4.3.2, we have defined the log-likelihood estimator:

$$\log \dot{f}_m := \log \hat{f}_m + \frac{\hat{\alpha}_2}{2m\hat{f}_m^2}, \ \log \dot{S}_k := \log \hat{S}_k + \frac{\hat{\beta}_2}{2k\hat{S}_k^2}, \tag{26}$$

where $\alpha_2$, and $\hat{f}_m^2$ are the sample variance and sample mean of $\hat{f}_1(z_i)$; $\beta_2$, and $\hat{S}_k^2$ are the sample variance and sample mean of $\hat{S}_1(z_j)$. Here, we show how this extra term in $\log \dot{f}_m$ or $\log \dot{S}_m$ leads to the cancellation of the leading terms in the bias, e.g., $-\frac{\alpha_2}{2mf^2}$.

Proof: The $\frac{\hat{\alpha}_2}{\hat{f}_m^2}$ is considered as a function of $g(x, y)$ in the form of $x/y^2$.

We expand its second-order Taylor expansion at $(\alpha_2, f)$:

$$\frac{\hat{\alpha}_2}{\hat{f}_m^2} = g(\alpha_2 + (\hat{\alpha}_2 - \alpha_2), f + (\hat{f}_m - f)) \approx \frac{\alpha_2}{f^2} + \frac{1}{f^2}(\hat{\alpha}_2 - \alpha_2) - \frac{2\alpha_2}{f^3}(\hat{f}_m - f) - \frac{2}{f^3}(\hat{\alpha}_2 - \alpha_2)(\hat{f}_m - f) + \frac{6\alpha_2}{2f^4}(\hat{f}_m - f)^2.$$

Notice that $\mathbb{E}[\hat{f}_m] = f; \mathbb{E}[\hat{\alpha}_2] = \alpha_2$. Taking expectation on both sides and after the rearrangement, we have

$$\mathbb{E}\left[\frac{\hat{\alpha}_2}{\hat{f}_m^2}\right] \approx \frac{\alpha_2}{f^2} - \frac{2}{f^3}\mathbb{E}[(\hat{\alpha}_2 - \alpha_2)(\hat{f}_m - f)] + \frac{3\alpha_2}{f^4}\mathbb{E}[(\hat{f}_m - f)^2].$$

Using the results in Zhang (2007), we have $\mathbb{E}[(\hat{\alpha}_2 - \alpha_2)(\hat{f}_m - f)] = \alpha_3/m$ and $\mathbb{E}[(\hat{f}_m - f)^2] = \alpha_2/m$. Thus, by substituting,

$$\mathbb{E}\left[\frac{\hat{\alpha}_2}{\hat{f}_m^2}\right] = \frac{\alpha_2}{f^2} - \frac{1}{m}(\frac{2\alpha_3}{f^3} - \frac{3\alpha_2^2}{f^4}) + o(m^{-1}).$$

Finally,

$$\begin{aligned}
\log f - \mathbb{E}[\log \dot{f}_m] &= \log f - \mathbb{E}[\log \hat{f}_m] - \frac{1}{2m}\mathbb{E}\left[\frac{\hat{\alpha}_2}{\hat{f}_m^2}\right] \\
&= \frac{1}{m}\frac{\alpha_2}{2f^2} - \frac{1}{m^2}(\frac{\alpha_3}{3f^3} - \frac{3\alpha_2^2}{4f^4}) - \frac{1}{2m}[\frac{\alpha_2}{f^2} - \frac{1}{m}(\frac{2\alpha_3}{f^3} - \frac{3\alpha_2^2}{f^4})] + o(m^{-2}) \\
&= \frac{1}{m^2}(\frac{2\alpha_3}{3f^3} - \frac{3\alpha_2^2}{4f^4}) + o(m^{-2}).
\end{aligned}$$

By applying the same proof as for $\mathbb{E}[\log \hat{S}_k]$, and $\dot{L}_{m,k}$, we complete the whole proof.

# C. Details of Experiments

### C.1. Experimental Setup

The experiments are on Python 3.9 with Pytorch on the Windows 11 system. GPU is not required.

### C.2. Simulated dataset (SD1-SD6)

The dimension of $x, y, u, c$ is 1. The latent dimension of $\boldsymbol{z} = (z_1, z_2)$ is 2. The Gibbs sampling process is designed as follows:

First, in the $P(X, Y, I|Z)$ step, we have a sample of $X, U, C, Y, I$ as follows

- The prior of x: $p(x) \sim N(1, 1)$, which is independent of $Z$.

- Given $Z$, $P(U|X, Z) \sim N(\mu(x, z), \sigma^2)$, where $\mu(x, \boldsymbol{z}) = 1 * z_1 + x * z_2$.

- Given $Z$, $P(C|X, Z) \sim N(\mu_C, e^2)$. The mean for SD1-SD6 is reported in the Table 1, which controls the rate of censoring.

- We compare the sampled $u, c$ to get $y$ and the event indicator $\delta$.

Second, in the $P(Z|X, Y, I)$ step, we define the distribution as follows:

- For $\delta = 0, 1$, $P(Z|X, Y, \delta)$ is normal distributed with mean $\mu_z = (2\delta - 1)(3/\exp(x + y), 3/\exp(x + y))$.

- Covariance is fixed as identity matrix for both cases.

Then we start the simulation at $z = (0, 0)$ and burn the first 10k observations.

### C.3. Hyper-parameters of training CD-CVAE and the variants

The details of the models can be found in the model folder via the repository link. In model specification, we have tuned the following hyper-parameters:

- Distribution Family of decoders: we choose from normal or gumbel-minimum.

- Network structure: the size of encoder and decoder networks and their depth.

- Dropout: the probability of dropout in the last layer of both encoder and decoders network. We select it from {0,0.2, 0.5, 0.9}.

- Latent dimension: the dimension of Z: we select from $2 * dim(x)$ or $0.5 * dim(x)$.

- For the variants with importance sampling, we set $m = k$ and choose it from {10,30,100}.

In the training stage, we have tuned the following hyper-parameters:

- Learning rate: 0.01, 0.001.

- batch size: 20, 100, 250, 500, 1000.

- Patience: the maximum number of epoch waiting until we stop the algorithm if no better validated metric is found. This helps reduce training time on overfitting the model.

- Temperature: reweighting parameter for the loss of censored observation, as introduced in Deep survival machine (Nagpal et al., 2021a). We choose 1, 1.3 or 0.9.

**C.4. Details in training-validating-testing stages of the experiments**

For the simulation dataset and inference gap in Table 2.

- We set the hyper-parameter $m = k = 10$ for IS and DVI variants.

- No validation and testing,since we know the truth. Best metric throughout the training process is reported.

- We use a Normal family for the decoder that aligns with the truth. Encoder/Decoder network shares the same network structure. Technical or adhoc hyper-parameters are avoided, e.g., temperature is set at 1, dropout is 0.

For the evaluation experiments on $C/C^{td}/$Brs in Table 4.

- Train-validation-test split ratio is 0.6, 0.2, 0.2. Experiment repetition is 5, using the same seeds of dataset split.

- Best model is selected from best cross-validated $C$ index or Brier score of the model taking on quantiles of survival time, predicting from validation $x$. We select it for a overall good fitting of the model, which is not the best validated metrics $C^{td}$ and Brier Score are valuated at specific test times to prevent overfitting.

- The hyper-parameters tuned for training SOTA models in the training stage follows the recommendations from Nagpal et al. (2022). For details, please refer to the package website or the source codes attached.

For the implementation of metrics, we note that

- $C$ index is implemented via Python package Pycox by the authors of Kvamme et al. (2019)

- $C^{td}$ and Brier score is implemented via Python package Scikit-Survival (Pölsterl, 2020).

## C.5. LogSumExp Trick of Computing Bias Term in Delta Method CDVI: A Step-by-Step Derivation

In this section, we introduce a numerically stable approach to calculate the bias terms in Delta Method Variational Inference (Teh et al., 2006) that is used in the delta method (DVI) variant of CD-CVAE. Inspired by the normalized importance weighting technique of Burda et al. (2015), our approach leverages a Log-Sum-Exp formulation that implicitly avoids over/underflow problems, which offers a more stable computation for gradient-based optimization.

Take Eq.26 above as an example, denoting $\hat{f}_1(z_i)$ as $w_i$:

$$\text{bias term} = \frac{1}{2m} \cdot \frac{\hat{\alpha_2}}{\hat{f}_m^2} \tag{27}$$

where $\hat{f}_m = \frac{1}{m}\sum_{i=1}^m w_i = \mathbb{E}[w], \alpha_2 = \frac{1}{m-1}\sum_{i=1}^k (w_i - \mathbb{E}[w])^2 = \mathbb{V}(w)$.

Goal: reformulate Eq.27 by **the logarithm of a normalized weight vector**, avoiding overflow during exponentiation.

$$\log \tilde{w}_i = \log \frac{w_i}{\sum_{j=1}^k w_j}.$$

Step 1: observe that $w_i - \mathbb{E}[w] = \tilde{w}_i \cdot \sum_j w_j - \mathbb{E}[w] = \tilde{w}_i \cdot m\mathbb{E}[w] - \mathbb{E}[w] = \mathbb{E}[w](m\tilde{w}_i - 1)$. Thus, plug into the sample variance:

$$\mathbb{V}(w) = \frac{1}{m-1}\sum_{i=1}^m (w_i - \mathbb{E}[w])^2 = \frac{1}{m-1}\sum_{i=1}^m [\mathbb{E}[w](m\tilde{w}_i - 1)]^2 = \frac{\mathbb{E}[w]^2}{m-1}\sum_{i=1}^m (m\tilde{w}_i - 1)^2$$

Step 2: substituting the above equation into the bias formula:

$$\text{bias term} = \frac{1}{2m} \cdot \frac{\mathbb{E}[w]^2}{\mathbb{E}[w]^2} \cdot \frac{1}{m-1}\sum_{i=1}^m (m\tilde{w}_i - 1)^2 = \frac{1}{2m(m-1)}\sum_{i=1}^m (m\tilde{w}_i - 1)^2$$

Since $(m\tilde{w}_i - 1)^2 = m^2\tilde{w}_i^2 - 2m\tilde{w}_i + 1; \sum_{i=1}^m \tilde{w}_i = 1$, summing over all $i \in \{1,\dots,m\}$:

$$\sum_{i=1}^m (m \cdot \tilde{w}_i - 1)^2 = \sum_{i=1}^m \left(m^2\tilde{w}_i^2 - 2m\tilde{w}_i + 1\right) = m^2\sum_{i=1}^m \tilde{w}_i^2 - 2m\sum_{i=1}^m \tilde{w}_i + m = m^2\sum_{i=1}^m \tilde{w}_i^2 - m.$$

Thus, the delta method bias term **for the density** becomes:

$$\text{bias term} = \frac{1}{2(m-1)}\left(m\sum_{i=1}^m \tilde{w}_i^2 - 1\right).$$

Step 3: Compute $\sum_{i=1}^m \tilde{w}_i^2$ via $\text{sum}\exp(\log \tilde{w}_i))$ or $\exp(\log \text{sum}\exp(\log \tilde{w}_i))$ safely with 1-2 exponential operations using the Log-Sum-Exp trick.

(Recommended) Step 4: Compute the same bias term of the log survival function, since the bias term must be computed separately for censored observations. (The bias terms in survival function estimates are generally well-behaved and bounded.)

# D. Additional Experiments

## D.1. Additional Experiments on Simulated Datasets

Table 7: Experiments are repeated 5 times with a $0.6, 0.2, 0.2$ train-validation-test split ratio. The best model is selected using cross-validated $C$-index. Test $C$-index are reported as mean $\pm$ standard deviation. Higher is better. Survival times for SD3–5 were transformed using the exponential function.

| Model | SD3 (20% cens) | SD4 (30% cens) | SD5 (50% cens) |
|---|---|---|---|
| **CoxPH** | $0.772 \pm 0.008$ | $0.739 \pm 0.005$ | $0.663 \pm 0.010$ |
| **SAVAE** | $0.765 \pm 0.012$ | $0.722 \pm 0.013$ | $0.653 \pm 0.014$ |
| **DeepSurv** | $0.788 \pm 0.010$ | $0.748 \pm 0.010$ | $0.657 \pm 0.008$ |
| **DSM** | $0.777 \pm 0.009$ | $0.742 \pm 0.011$ | $0.653 \pm 0.005$ |
| **RSF** | $0.789 \pm 0.009$ | $0.756 \pm 0.008$ | $\mathbf{0.677} \pm 0.007$ |
| **DCM** | $0.774 \pm 0.010$ | $0.752 \pm 0.008$ | $0.675 \pm 0.007$ |
| **CD-CVAE** | $0.789 \pm 0.013$ | $\mathbf{0.758} \pm 0.014$ | $0.673 \pm 0.012$ |
| **CD-CVAE$^{+\mathbf{IS}}$** | $\mathbf{0.791} \pm 0.015$ | $0.761 \pm 0.012$ | $0.662 \pm 0.010$ |
| **CD-CVAE$^{+\mathbf{DVI}}$** | $0.772 \pm 0.009$ | $0.741 \pm 0.008$ | $0.665 \pm 0.008$ |

## D.2. Additional Experiments on Real-World Datasets

Table 8: Experiments are repeated five times with a $0.6, 0.2, 0.2$ train-validation-test split ratio. Test $C$-index are reported as mean $\pm$ standard deviation. Higher is better. Bold entries indicate the highest mean per column(dataset).

| Model | SUPPORT | FLCHAIN | NWTCO | METABRIC | WHAS | GBSG | PBC |
|---|---|---|---|---|---|---|---|
| CoxPH | $0.661 \pm 0.006$ | $0.623 \pm 0.207$ | $0.685 \pm 0.009$ | $0.637 \pm 0.009$ | $0.756 \pm 0.018$ | $0.675 \pm 0.006$ | $0.829 \pm 0.014$ |
| SAVAE | $0.625 \pm 0.008$ | $0.745 \pm 0.023$ | $0.689 \pm 0.018$ | $0.643 \pm 0.020$ | $0.751 \pm 0.022$ | $0.668 \pm 0.010$ | $0.811 \pm 0.028$ |
| DeepSurv | $0.644 \pm 0.005$ | $0.775 \pm 0.005$ | $0.670 \pm 0.019$ | $0.640 \pm 0.019$ | $0.758 \pm 0.021$ | $0.597 \pm 0.011$ | $0.828 \pm 0.017$ |
| DSM | $0.668 \pm 0.008$ | $0.782 \pm 0.010$ | $0.691 \pm 0.024$ | $0.673 \pm 0.010$ | $0.789 \pm 0.022$ | $0.618 \pm 0.017$ | $0.836 \pm 0.018$ |
| RSF | $0.677 \pm 0.005$ | $0.762 \pm 0.015$ | $0.683 \pm 0.013$ | **$0.679 \pm 0.005$** | $0.791 \pm 0.015$ | **$0.685 \pm 0.012$** | $0.841 \pm 0.018$ |
| DCM | $0.676 \pm 0.004$ | $0.785 \pm 0.003$ | $0.678 \pm 0.003$ | $0.670 \pm 0.079$ | $0.779 \pm 0.015$ | $0.604 \pm 0.014$ | $0.841 \pm 0.018$ |
| CD-CVAE | **$0.679 \pm 0.003$** | **$0.794 \pm 0.012$** | $0.690 \pm 0.018$ | $0.678 \pm 0.012$ | **$0.842 \pm 0.026$** | $0.678 \pm 0.025$ | **$0.845 \pm 0.011$** |
| CD-CVAE$^{+\text{IS}}$ | $0.672 \pm 0.007$ | $0.787 \pm 0.012$ | **$0.706 \pm 0.012$** | $0.661 \pm 0.015$ | $0.822 \pm 0.019$ | $0.663 \pm 0.012$ | $0.839 \pm 0.010$ |
| CD-CVAE$^{+\text{DVI}}$ | $0.670 \pm 0.004$ | $0.791 \pm 0.013$ | $0.671 \pm 0.025$ | $0.669 \pm 0.015$ | $0.814 \pm 0.014$ | $0.675 \pm 0.011$ | $0.828 \pm 0.016$ |

Table 9: Experiments are repeated five times with a $0.6, 0.2, 0.2$ train-validation-test split ratio. Test $C^{td}$ are reported as mean $\pm$ standard deviation. Higher is better. Bold entries indicate the highest mean per column(dataset).

| Model | SUPPORT | FLCHAIN | NWTCO | METABRIC | WHAS | GBSG | PBC |
|---|---|---|---|---|---|---|---|
| CoxPH | $0.664 \pm 0.006$ | $0.626 \pm 0.202$ | $0.688 \pm 0.010$ | $0.637 \pm 0.009$ | $0.761 \pm 0.016$ | $0.676 \pm 0.004$ | $0.831 \pm 0.012$ |
| SAVAE | $0.631 \pm 0.012$ | $0.773 \pm 0.016$ | $0.675 \pm 0.023$ | $0.657 \pm 0.012$ | $0.770 \pm 0.012$ | $0.655 \pm 0.018$ | $0.815 \pm 0.021$ |
| DeepSurv | $0.644 \pm 0.005$ | **$0.795 \pm 0.012$** | $0.721 \pm 0.018$ | $0.656 \pm 0.015$ | $0.742 \pm 0.021$ | $0.602 \pm 0.015$ | $0.842 \pm 0.011$ |
| DSM | $0.668 \pm 0.008$ | $0.789 \pm 0.007$ | $0.683 \pm 0.011$ | $0.660 \pm 0.010$ | **$0.805 \pm 0.014$** | $0.653 \pm 0.011$ | $0.858 \pm 0.011$ |
| RSF | $0.651 \pm 0.005$ | $0.782 \pm 0.015$ | $0.713 \pm 0.013$ | **$0.680 \pm 0.005$** | $0.802 \pm 0.013$ | **$0.718 \pm 0.010$** | **$0.859 \pm 0.009$** |
| DCM | $0.672 \pm 0.006$ | $0.793 \pm 0.010$ | $0.723 \pm 0.014$ | $0.679 \pm 0.014$ | $0.801 \pm 0.012$ | $0.624 \pm 0.013$ | $0.858 \pm 0.012$ |
| CD-CVAE | $0.672 \pm 0.005$ | $0.789 \pm 0.015$ | **$0.741 \pm 0.012$** | $0.660 \pm 0.022$ | $0.802 \pm 0.013$ | $0.692 \pm 0.015$ | $0.845 \pm 0.018$ |
| CD-CVAE$^{+\text{IS}}$ | **$0.677 \pm 0.006$** | $0.756 \pm 0.022$ | $0.717 \pm 0.022$ | $0.655 \pm 0.012$ | $0.796 \pm 0.015$ | $0.663 \pm 0.012$ | **$0.859 \pm 0.012$** |
| CD-CVAE$^{+\text{DVI}}$ | $0.663 \pm 0.004$ | $0.751 \pm 0.014$ | $0.709 \pm 0.017$ | $0.659 \pm 0.008$ | $0.788 \pm 0.012$ | $0.688 \pm 0.017$ | $0.858 \pm 0.012$ |

Table 10: Experiments are repeated five times with a $0.6, 0.2, 0.2$ train-validation-test split ratio. Test Brier Scores are reported as mean $\pm$ standard deviation. Lower is better. Bold entries indicate the lowest mean per column(dataset).

| Model | SUPPORT | FLCHAIN | NWTCO | METABRIC | WHAS | GBSG | PBC |
|---|---|---|---|---|---|---|---|
| CoxPH | $0.218 \pm 0.002$ | $0.146 \pm 0.058$ | $0.099 \pm 0.004$ | $0.216 \pm 0.004$ | $0.175 \pm 0.011$ | $0.225 \pm 0.008$ | $0.126 \pm 0.010$ |
| SAVAE | $0.224 \pm 0.005$ | $0.122 \pm 0.006$ | $0.117 \pm 0.010$ | $0.218 \pm 0.012$ | $0.169 \pm 0.012$ | $0.224 \pm 0.005$ | $0.138 \pm 0.011$ |
| DeepSurv | $0.215 \pm 0.008$ | $0.117 \pm 0.005$ | $0.083 \pm 0.008$ | $0.235 \pm 0.006$ | $0.209 \pm 0.010$ | $0.245 \pm 0.005$ | $0.135 \pm 0.007$ |
| DSM | $0.240 \pm 0.012$ | $0.121 \pm 0.007$ | $0.084 \pm 0.011$ | $0.235 \pm 0.008$ | $0.181 \pm 0.006$ | $0.248 \pm 0.006$ | $0.136 \pm 0.008$ |
| RSF | $0.225 \pm 0.001$ | $0.127 \pm 0.006$ | $0.088 \pm 0.007$ | $0.221 \pm 0.002$ | **$0.166 \pm 0.006$** | **$0.223 \pm 0.007$** | **$0.123 \pm 0.005$** |
| DCM | $0.220 \pm 0.004$ | $0.118 \pm 0.006$ | **$0.081 \pm 0.007$** | $0.221 \pm 0.006$ | $0.175 \pm 0.006$ | $0.233 \pm 0.006$ | $0.141 \pm 0.005$ |
| CD-CVAE | $0.218 \pm 0.005$ | **$0.115 \pm 0.005$** | **$0.081 \pm 0.006$** | **$0.211 \pm 0.009$** | $0.178 \pm 0.007$ | $0.234 \pm 0.005$ | $0.127 \pm 0.004$ |
| CD-CVAE$^{+\text{IS}}$ | **$0.211 \pm 0.006$** | $0.122 \pm 0.005$ | $0.093 \pm 0.010$ | $0.222 \pm 0.012$ | $0.172 \pm 0.005$ | $0.235 \pm 0.005$ | $0.144 \pm 0.010$ |
| CD-CVAE$^{+\text{DVI}}$ | $0.228 \pm 0.005$ | $0.125 \pm 0.004$ | $0.100 \pm 0.005$ | $0.226 \pm 0.006$ | $0.169 \pm 0.022$ | $0.224 \pm 0.005$ | $0.138 \pm 0.011$ |

