# OpenReview forum: "Censor Dependent Variational Inference"
_ICML.cc/2025/Conference — ICML 2025 poster_

### Official Review · Reviewer_KuLt · 2025-03-14

**Overall Recommendation:** 3

**Summary:**

The paper proposes a censor-dependent conditional VAE (CD-VAE), where two variational posteriors—for censored and non-censored events—are inferred given covariates and observed times, instead of the typical single posterior assumed in baseline methods. Further, the paper provides theoretical results to support the decomposed posterior. Experimental results on synthetic datasets show (i) smaller KL divergence between the prior and posterior ;  (ii) competitive performance in terms of the C-Index and Brier score.

**Claims And Evidence:**

- The paper claims significant performance improvements over baselines:

1) In general, the experimental results demonstrate marginal or comparable performance compared to baselines. It's unclear whether the marginal gains are statistically significant, as confidence intervals are not provided.

2) It's unclear why C-Index and Brier Score results are not reported on the synthetic datasets, where only KL divergence is provided. This could be misleading, as a small KL divergence between the prior and posterior could also imply posterior collapse and is not necessarily indicative of survival model performance.

3) It's unclear why the paper does not benchmark against VSI (Xiu et al., 2020), which proposes a principled, single shared posterior for censored and non-censored approaches on real-world datasets (Table 4).

4) It's unclear why the real-world experiments (Table 4) do not include the importance sampling variations of the proposed CD-VAE approach.

**Essential References Not Discussed:**

- The paper should discuss other variational approaches outside survival analysis focusing on *conditional* posterior distributions, including Siddharth et al. (2017), Kingma et al. (2014), and Joy et al. (2021).

**Experimental Designs Or Analyses:**

- See methods and evaluation section.

**Methods And Evaluation Criteria:**

- The decomposed posterior seems problematic, as only the observed time depends on the censoring indicator, which is already accounted for in the log-likelihood. This also reduces the number of samples used to learn the inference, where the quality of the learned posterior becomes a function of the censoring rate. I encourage the authors to provide complete results (C-Index, Brier score, KL divergence, and calibration) on both synthetic and real-world datasets to comprehensively evaluate the effect of censoring on these metrics. Additionally, including the censoring rate in Table 3 would be more informative than simply reporting the number of censored samples.

- The paper should also consider evaluating the approach on larger datasets with more covariates, such as the SEER dataset, to provide insights into how the proposed method scales with the number of covariates and sample size.

- The paper states that the Brier score is a measure of calibration, which is not necessarily true. See Haider et al. (2020) for D-calibration and Chapfuwa et al. (2020) for KM-calibration as metrics for survival calibration.

**Other Comments Or Suggestions:**

- Given that Theorems 4.3.1–4.3.3 are straightforward extensions of previously proposed theorems and do not appear to be directly related to the key contribution on the decomposed censor-dependent posterior, it is unclear whether they should be included in the main paper. The writing could be improved to clarify the connections; otherwise, it is not clear what purpose the proposed theorems serve in strengthening the key contributions.

**Minor**
- Remove empty [] from Brier score equation

**Other Strengths And Weaknesses:**

- A major strength of this paper lies in its connections to previously proposed variational survival analysis methods and variational inference. However, the justification of the proposed decomposed censor-dependent variational distribution and its practical implications—particularly the sensitivity to the censoring rate, which has not been thoroughly explored in the paper—represent a major weakness.

**Questions For Authors:**

- Could you provide comprehensive results on synthetic and real-world datasets, including the impact of the censoring rate on the learned posteriors?
- Could you clarify why, in Lemma 3.1, the equivalence between equations (4) and (5) is necessary for *non-censored* events, given that only equation (4) is used in the likelihood estimation in equation (6) when events are *non-censored*?

**Relation To Broader Scientific Literature:**

- Variational inference is an important generative modeling approach, where most methodological contributions focus on the posterior/prior distribution. This paper focuses on justifying the proposed decomposed censor-dependent posterior for variational survival analysis.

**Theoretical Claims:**

- Lemma 3.1: Justifying the need to decompose the posterior in Theorem 3.2.1 could be problematic:

1) The paper begins by establishing conditions of equality for equations (4) and (5) assuming *non-censored* events. However, it is unclear why it is necessary for these equations to be equal, as the likelihood for observed events is already decomposed in equation (6) according to the censoring indicator. Moreover, note the relationship  $f(t|x) = h(t|x) \cdot S(t|x)$, which implies that (4) and (5) are equal only if the hazard function is constant at 1.

2) Other theoretical claims appear to be straightforward extensions of previously proposed theorems.

---

> ### Author Rebuttal · Authors · 2025-03-28
>
> We sincerely thank the reviewer for their time and effort in engaging with our work. Your recognition of the importance of variational survival analysis is encouraging. We've added further experimental results and endeavor to clarify a few points that may have been misinterpreted.
>
> ## Response to Experimental Concerns
> See extended experimental results in response to Reviewer acdJ (#1).
>
> >Evidence[2]: reporting C-index on the synthetic datasets
>
> Our experiments aim to evaluate both the inference optimality of our method and its effectiveness in survival modeling. Our simulation datasets are essential for assessing the former. Nonetheless, we’ve included additional results for completeness.
>
> >Evidence[1,3,4]:significance metrics; comparison with VSI. report variants on benchmark
> >Methods[2]:SEER datasets
>
> Statistical significance testing is not a common practice. VSI lacks a training script. SEER used in Nagpal et al. (2021a) is against our modeling assumptions as a dataset for competing risks analysis.
>
> In response, we added the Apellaniz et al. (2024) baseline, reported average metrics with uncertainties, and included full performance results for all variants.
> >Evidence[2]:misleading KLD
>
> >Weaknesses: lack of sensitivity exploration
>
> We would like to clarify a misunderstanding: the KL divergence reported in Table 2 is with respect to the true posterior, not the prior, as noted in the caption. Simulated datasets were used to explore the optimality of CDVI under varying censoring rates, while the seven benchmark datasets naturally cover a wide range of censoring scenarios.
>
> ## Addressing Theoretical Questions
> 1. Methods[1] & Weekness[1]: A potential issue with the proposed variational distribution is the assumption that the observed time Y depends on the censoring indicator δ, which lacks theoretical justification.
> 2. Theoretical Claims[1] & Questions[2]: Justifying the need to decompose the posterior in Theorem 3.2.1 could be problematic: Why necessary for Eq4 and Eq5 to be equal?
> 3. Methods[1] & Other Weaknesses[1]: Separating network parameters reduces the number of samples used to learn the inference and can introduce sensitivity to the censoring rate.
>
> We appreciate the opportunity to clarify all these points regarding our theory:
> 1. The dependency between Y and δ is not assumed in the posterior. See also Reviewer oybt (#2)'s first question.
> 2. Eq. 4 and 5 are not required to be equal. They are inequalities that can be required to hold equal on both sides simutanuously on a same data point (x,y). See Claims [1] in Reviewer acdJ (#1) for a complete reasoning.
> 3. Variational parameter separation is not forced or by default — see our reply to Reviewer acdJ (#1). We use a joint encoder and our proposed variational posterior approximates the true posterior $p(z|x,y,\delta)$ as a whole; thus, sample imbalance is a general issue in (variational) conditional density estimation, not unique to our method; we address it following standard strategies (Nagpal et al., 2021a, Line 1042).
>
> We hope these clarifications resolve the concerns and misunderstandings that you mentioned as a major weakness. Nonetheless, we remain happy to provide further detail if helpful!
>
> >Suggestions[1]: Importance of Theorems 4.3.1–4.3.3
>
> Thank you for the comments. While these results build on existing work, our analysis are concise—occupying less than half a page—and are carefully presented to support, rather than distract from, the central idea of censor-dependent variational inference (CDVI). They offer additional theoretical justification for our proposed CDVI by demonstrating its alignment with the broader augmented VI framework. They also offer rigorous foundations for our proposed variants, offering theoretical grounding of design choices that has not been established in prior work.
>
> Importantly, extending these results to the survival setting required nontrivial technical work. As mentioned in line 300, Theorem 4.3.2 corrects a key issue in the original framework by identifying a notational inconsistency between the variational posterior and the augmented variational posterior. To our knowledge, this inconsistency has not been previously addressed in the literature. As a result, Theorem 2 from Domke & Sheldon (2018) could not be directly extended to our setting. For reference, we invite the reviewer to compare our proof in Appendix B.6 with the proof of Theorem 1 in Domke & Sheldon (2018).
> > Brier score is not necessarily a calibration;  Adding References.
>
> Thank you for highlighting this extremely helpful detail. We have revised the interpretation in line with Haider et al. and added the mentioned references along with [Sohn et al. 2015] for conditional variational posterior/VAEs.
>
> *We thank the reviewer again for the valuable feedback. Given the clarifications and efforts made, we would appreciate your consideration of reassessing the score. Please don’t hesitate to reach out with further questions.

---

> > ### Comment · Reviewer_KuLt · 2025-04-03
> >
> > Thank you for addressing some of my concerns. I have read the reviews and responses, as well as the justifications for the proposed CDVI. However, it is still unclear how the proposed CDVI ELBO in Eq. (12) is better than the vanilla VSI ELBO.
> >
> > - **The Vannial VSI  ELBO:**
> > $\text{ELBO}(x|t) = \mathbb{E}_{q(z|t)} [\log p(t|z)] - KL( q(z|x, t) || p(z|x) )$
> >
> > **Proposed CDVI ELBO Eq. (12):**
> > By decoupling the learned posteriors for censored and observed events, CDVI reduces the number of samples required for learning the inference process. Consequently, the quality of the learned posteriors becomes dependent on the censoring rate. It is unclear how to assess this key limitation without comprehensive analysis and direct comparison with VSI. The VSI GitHub repository is available here: https://github.com/ZidiXiu/VSI/.

---

> > > ### Author Response · Authors · 2025-04-04
> > >
> > > We truly appreciate your time and efforts in reading our rebuttal. We are grateful for your acknowledgement and expressing the remaining concerns.
> > > >  [Q1] still unclear how the proposed ELBO is better than the vanilla ELBO (citing Eq. 4 in [Xiu, 2020]).
> > >
> > > Thank you for the additional question. However, we would like to kindly clarify a misunderstanding: Eq.4 cited from [Xiu, 2020](https://arxiv.org/pdf/2003.04430) is **not** their proposed ELBO. As noted in the title of Section 3.1 in Xiu, 2020, Eq. 4 corresponds to the variational bound for *observed events* only. Their proposed ELBO is obtained by substituting terms in Eq.7 with Eq.4 and the bound for *censored events* in Section 3.2, corresponding to the right-hand sides of our Eq. 4 and Eq. 5 (Lines 99–100). For correctness, we continue to refer to our **Eq. 6** as vanilla ELBO, which consistent with the derivation in Xiu, 2020 and the derivation in [Nagpal 2021a, page 4](https://arxiv.org/pdf/2003.01176).
> > >
> > > Our analysis in Section 3 demonstrates why and how Eq.12 is preferred for a better inference optimality.
> > > - As noted in Section 2.4, VI optimality is achieved when the variational posterior (encoder) closes the inference gap between the log-likelihood in Eq. 2 and the ELBO in Eq. 6 or Eq. 12.
> > > - The line of argument in Lemma 3.1->Prop. 3.1->Remark 3.1 proves that this inference gap cannot be properly tightened using a vanilla $q_\phi(z|x,y)$ and the derived vanilla ELBO in Eq.6: an optimal $q_\phi$ vanilla VI solution fails to exist for non-degenerate θ (claim 1-2) or may be a trivially learned solution exhibiting laziness or collapse (claim 3-4). These issues are triggered when the presence of both (x,y,1) and (x,y,0) data triplets, which are unavoidable when their corresponding supports overlap.
> > > - The line of argument in Thm 3.2.1->Remark 3.2->Thm 3.2.2 proves that the optimal encoder for log-likelihood in Eq.2 that avoids the above issues should be designed to approximate to the true posterior density $p(z|x,y,\delta)$ *as a whole*, thereby avoiding the separating variational bounds in Eq. 4 and Eq. 5, as done in [Xiu, 2020] or [Nagpal 2021a]. Our proof of Thm 3.2.1 in Appendix B.2 shows that this requires to properly incorporate the censoring indicator $\delta$. As a result, Thm 3.2.2 proves how CDVI benefits from such $q_\phi(z|x,y,\delta)$ as it avoids the hard constraint $\phi_1=\phi_2$ imposed by the vanilla variational posterior q(z|x,y).
> > >
> > > In practice, as shown in Table 2, using the vanilla ELBO in Eq. 6 within a v-structured CVAE results in a large inference gap on simulation dataset SD2-5, which is substantially reduced by the censor-dependent ELBO in Eq. 12 (and its sampling variants). Extensive validation on benchmark datasets also shows a better estimation of individual survival distributions over multiple methods DSM, SAVAE based on vanilla ELBO.
> > > > Key Limitation: By decoupling the learned posteriors for censored and observed events, CDVI reduces the number of samples required for learning the inference process. Consequently, the quality of the learned posteriors becomes dependent on the censoring rate.
> > >
> > > Thank you for the comments. However, we would like to clarify that this misunderstanding does not align with our methodology, which has already been addressed in our first rebuttal and response to the question from Reviewer acdJ(#1). Specifically, the encoder in Eq. 10 is implemented as a dense, joint network to approximate the true posterior density $p(z|x,y,\delta)$ *as a whole*, as shown in Fig.3(a). An awareness of parameter sharing and network partitioning provides clarity on why $\phi_1$ is not trained separately on all event observations.
> > >
> > > Additionally, we clarify that the implication introduced by “consequently, .....” is misleading. In general, performance degradation as the censoring rate increases is a well-known challenge and not specific to our approach (or more broadly variational methods); see also Table 1 from Xiu, 2020. Similarly, as evident in Table 2, the accuracy of the learned posteriors in CVAEs varies with the censoring rate using both vanilla ELBO and our proposed ELBO. However, our proposed method consistently improves the accuracy of the learned posteriors in SD2-5 datasets with censoring rate ranging from 5%-50% compared to a vanilla CVAE that uses a vanilla ELBO.
> > >
> > > **Edited on 4/7: We respectfully follow up to inquire if any feedback is available on this point. As Reviewer #1 suggested, we have included an illustrative figure (https://imgur.com/a/1BtArNl) to show why censor dependence is essential for modeling true posterior under censoring.**
> > > > A direct comparison with VSI.
> > >
> > > Given that vanilla EBLO has been extensively compared in VAE-based DSM, SAVAE(Apellániz et al. 2024) and our CVAE implementations, we believe the current evaluations sufficiently highlight the strengths of our method, particularly as VSI’s training script is not yet publicly available (syntax to run VSI is still under contribution).

---

### Official Review · Reviewer_oybt · 2025-03-17

**Overall Recommendation:** 3

**Summary:**

This paper builds upon prior work (Nagpal et al. 2021a, Apellaniz et al. 2024) that uses the variational distribution $q_\phi (z \mid x, y)$ as a posterior approximation for $p_\theta (z\mid x)$, without accounting for censoring (i.e., $y$ and $\delta$). The authors propose a variational inference method that explicitly incorporates censoring, providing a tighter bound on the log-likelihood. The proposed approach is validated through extensive experiments on six simulation studies and seven real-world datasets, demonstrating empirical effectiveness.

## Update after rebuttal
Thank you for the detailed response and for conducting the additional experiments — they address most of my concerns.

Additionally, I realize my earlier statement, "using the predicted probability as score is problematic," may have been unclear. What I intended to convey is that referring to the current version of the concordance index as "Harrell’s C-index" can be misleading, as it differs from the formulation originally described in Harrell’s paper.

**Claims And Evidence:**

The theoretical contributions and experimental results strongly support the paper’s claims.

**Essential References Not Discussed:**

No.

**Experimental Designs Or Analyses:**

Yes. I've checked the experimental design.

However, there are two issues:
* The Harrell's concordance index this paper claims using is problematic. The correct way of calculating Harrell's C-index is by comparing the predicted times, not the predicted probability at a single time. Harrell's paper ([link](https://onlinelibrary.wiley.com/doi/10.1002/(SICI)1097-0258(19960229)15:4%3C361::AID-SIM168%3E3.0.CO;2-4), Section 5.5) clearly states that the predicted survival time is the default choice and the predicted probability can serve as a substitute with **conditions**. This substitution is only allowed when the predicted probabilities and times have a one-to-one mapping (e.g., proportional hazard is satisfied). However, in the experiments, some of the baselines (RSF, DSM) as well as the proposed method do not have the proportional hazard assumption and therefore using the predicted probability as the risk score is problematic and therefore not the correct Harrell's C-index.
* The reported performance is based on the highest value across five random seeds, which raises concerns about fairness. Variational inference methods are known to be challenging to optimize, as they heavily depend on good parameter initialization -- a well-documented issue in the literature. In my previous experiments, different random seeds significantly affected performance, leading to high variance. Selecting the best-performing result rather than reporting the average (or another robust statistic) gives an unfair advantage over more stable and robust methods.

**Methods And Evaluation Criteria:**

The overall methodological approach is reasonable—optimizing two distinct variational distributions for censored and event groups. However, I have two concerns:

* This approach seems to suggest a dependence between censoring times and event times, which contradicts the assumed DAG. The authors should clarify this apparent inconsistency.
* The paper only compares against one variational inference method (Nagpal et al. 2021a). Why are other relevant methods, such as Ranganath et al. (2016) and Apellaniz et al. (2024), excluded? A broader comparison would strengthen the evaluation.

**Other Comments Or Suggestions:**

* Line 90: "surjective" → "subjective"
* Lines 99, 126: "partial log-likelihood" → "log-likelihood" (The term partial log-likelihood in survival analysis specifically refers to Cox's proportional hazards model.)
* Figure 2: The meaning of solid vs. dashed lines is unclear. Suggestions: (1) make the caption self-contained, explicitly defining these lines.
and (2) use dashed lines for the encoder in Figure 3(a) to maintain consistency.
* Lines 352-353: "Deep Survival Forest (DSF)" → "Random Survival Forest (RSF)"
* The term *Censor-dependent* Conditional VAE suggests a dependent censoring assumption, which is misleading. A name like Censor-Aware VAE might be clearer.

**Other Strengths And Weaknesses:**

The paper is difficult to follow, particularly regarding the motivation and limitations of prior work. The introduction uses vague terms such as “remain unclear” without specifying concrete shortcomings.

**Questions For Authors:**

* In the general generative DAG of $U$ (Figure 1(a), Figure 2(b), and Figure 3(b)), why do you represent the sequential graph as $x \rightarrow z \rightarrow u$? Specifically, in your decoder, why is $x$ necessary? Additionally, how reasonable is the structure of the graph you are using?
* You mentioned that the event distribution should belong to a location-scale family. However, in your experiments, you did not explicitly specify which type of distribution you used for optimal modeling.

**Relation To Broader Scientific Literature:**

This paper extends previous work (Nagpal et al. 2021a, Apellaniz et al. 2024) by explicitly incorporating censoring into the variational inference framework.

**Theoretical Claims:**

I did not verify the correctness of the proofs in the appendix.

---

> ### Author Rebuttal · Authors · 2025-03-28
>
> We sincerely thank you for your thoughtful feedback and valuable suggestions. We truly appreciate the recognition of the contribution and your overall support of our work. We have improved our manuscript based on your recommendations. In what follows, we respond to your comments point by point, with deeper discussions where needed.
>
> > **A dependence between censoring times and event times contradicts the assumed DAG**
>
> This misconception is also raised by Reviewer KuLt (#3). The assumed DAG aligns with a conditional independence between uncensored time-to-event $U$ and censoring times $C$ as noted in line 93. A fact in probability theory is that the independence between $U$ and $C$ *does not* imply an independence between the **observed** censored times ${Y|\delta=0}$ and the **observed** events times ${Y|\delta=1}$ due to Eq.1.
> >VI relies on good parameter initialization; more experiements
>
> Thank you for sharing your insights. We found that tuning hyperparameters effectively reduces seed variability on datasets like NWTCO. Happy to share our logsumexp and xvaier tricks for stability or train script if needed. Ranganath et al. (2016) do not have a open script. The link for 4 additional experiments is in our reponse to Reviewer acdJ (#1).
> > **Computing of Harrell's C-index via predicted probabilities for non-cox models violates the one-to-one mapping condition**
>
> We appreciate the opportunity to having a deeper discussion regarding Harrell’s C-index. The issue you raised was carefully discussed during our research. We invite your additional feedback to our step-by-step reasoning below:
>
> 1.  We view the C-index, when computed using a particular risk score, as a generalized ranking metric/statistic that applies across all models considered. While Harrell’s C-index is defined using predicted survival times, the notion of concordance naturally extends to ranking via risk scores such as survival probabilities [Uno et al. 2011].
>
> 2. Proportional hazards (PH) assumption brings ordering guarantees. In particular, it ensures time-consistent ordering i.e., $S(u│x_i)≤S(u│x_j) ↔ ∀u, S(u|x_i)≤S(u│x_j)$ and risk-consistent ordering across different scores: $S(u│x_i)≤S(u│x_j) ↔ E(u|x_i)≤E(u│x_j)↔z(x_i)\geq z(x_j)$, which is refered to as one-to-one mapping [Harrell et al 1996]. The latter allows for efficient concordance computation via hazard ratios for Cox models, which is an advantage over models like RSF, DeepSurv, DSM, and our own. From this perspective, the averaged C-index offers a practical and robust measure of overall ranking performance.
>
> 3.  Without PH assumption, the inconsistency across *risk scores* is expected, and C-index based on a meaningful score remains valid and is not a "misuse"; the inconsistency across *test time* motivates the adoption of Antolini et al.’s [2005] in our work. Nevertheless, outperforming Cox models under both average and time-dependent C-indices demonstrates effective ranking without relying on the PH assumption.
>
> In conclusion, we disagree with the claim that "using the predicted probability as score is problematic" for non-Cox models
>
> > Clarification on the motivation and limitations of prior work
>
> Thank you for the feedback.  As discussed in Lines 142–150, our motivation stems from the lack of variational inference (VI) optimality analysis [Cremer et al., 2018] in prior work. The limitations in achieving VI optimality are shown in Prop. 3.1, while Remarks 3.1 and 3.2 point out issues in existing assumptions and model designs.
>
> > Why the sequential graph as x→z→u? Why x necessary? How reasonable is the v-structure?
>
> This is an insightful question—one we have also considered. To clarify, only Fig1(a) represents a general generative DAG; Fig2(b) and 3(b) depict the v-structured latent process used in our method. The sequential graph x→z→u corresponds to a D-separated latent structure, as in Ranganath et al. (2016) and Apellaniz et al. (2024), where $u$ depends solely on z. In contrast, the v-structure, assuming z⊥x, requires x to generate u, thereby enabling individual survival modeling.
>
> While both latent structures are equivalent [Zheng et al. 2022, Remark 1], the v-structure is the default setup of CVAE [Sohn et al. 2015, Section 3]. For survival analysis tasks, we conjecture that the absence of v-structure stems from the prevailing use of vanilla VI (since Ranganath et al. 2016), and its inferior VI optimality compared to D-separation as noted in Line 175-180.
>
> > Specify which type of location-scale family used for optimal modeling.
>
> As noted in Line 1023, it is treated as a hyperparameter with no consistent advantage observed; results depend on settings of  others hyperparameters.
>
> >Suggestions: Line 90: "surjective" → "subjective"
>
> It is not a typo. In mathematics, a function $f:A→B$ is surjective if for every b∈B, there exists at least one a∈A such that $f(a)=b$
>
> > Suggestions [2-4]
>
> Thank you for the valuable suggestions. We have revised the relevant section accordingly.

---

### Official Review · Reviewer_acdJ · 2025-03-17

**Overall Recommendation:** 3

**Summary:**

This paper analyzes the current practices to apply variational inference to latent variable models for survival analysis, provides insights into why the naive application of VI may be insufficient, and presents a new VI formulation that can potentially sidestep some of those challenges. The authors also include some experimental results that corroborate the improved performance from using the new framework.

**Claims And Evidence:**

Overall claims made at the start of the paper are well supported. I do have specific concerns and raise them in the subsequent sections.

**Essential References Not Discussed:**

I think references are missing from some important places in this paper. A few instances are
- The initial paragraphs of the introduction make claims about the applications of survival analysis.
- In lines 100-102, the authors are talking about a well-established way to define the likelihood in the survival analysis. It will be great to see some references here.
- In lines 175-180, there is discussion of posterior collapse without any referencing.
- In lines 195-205, the authors talk about different types of censoring without describing or referencing.
In general a pass over the paper may reveal even more places that would benefit from citations to provide support and additional context for readers not very well-read within the survival analysis literature.

**Experimental Designs Or Analyses:**

Please see the section above about the methods and evaluation.

**Methods And Evaluation Criteria:**

While the benchmarks and the metrics might be enough, I am confused about the lack uncertainty estimates in the numbers in Table 4 and Table 5. Can the authors comment on why they used the maximum and the minimum numbers across the trials without any uncertainty quantification?

**Other Comments Or Suggestions:**

Some typos and minor errors:
- $S_\theta$ in Eq. 2 started as $S_{U, \theta}$ and is then never used like that, again.
- $h$ is not defined before usage in 1) of Proposition 3.1
- $\pi$ in 4) of Proposition 3.1
- Lines 155-157, second column the wording is strange. I think the authors want to claim to be the first to identity the problem latent non-identifiability in survival analysis. Wording can be changed to make that more precise. Also, similarly, a more precise claim for correcting the result of Domke and Sheldon can be used in lines 301-303.
- Trailing bracket in equation for proposition 4.2
- What is DVI in the header part of section 5.

**Other Strengths And Weaknesses:**

Overall, I am want to accept this paper. However, I am can not recommend a clear accept in its current state. I think the paper is well written for most parts. But I also had trouble understanding the main take-aways from the analysis. Here are the few things that I think can help this paper a lot.
- Add references where needed to provided appropriate context
- Summarized the main results and take-aways from the analysis in the introduction with forward referencing.
- Add more context around the technical contributions in the Section 4. Also, please add forward referencing to proofs in the main body.
- Close the loop on the final algorithm for the proposed CD-CVAE approach with details about $S$.
- Motivate the main benefits of the proposed approach using a working example. I think a clear, simple demonstration of where the vanilla VI fails and where the proposed approach will succeed will go a long way in understanding the scope of the contributions.
- Add the proper uncertainty estimates in the empirical results.

**Questions For Authors:**

Please see the previous sections.

**Relation To Broader Scientific Literature:**

Survival analysis models are crucial in various scientific fields. As their applications increase, it becomes increasingly important to learn not only the models themselves but also the uncertainty associated with unobserved variables. Variational inference (VI) plays a pivotal role in addressing this challenge. The primary contribution of this work lies in theoretical analysis of what happens when naive VI is applied and how a more cautious approach can enhance its effectiveness. By providing essential foundations, this work paves the way for future research and development in mechanistic models for survival analysis.

**Theoretical Claims:**

Overall, I had trouble understanding the final take-away from the analysis. Here is what I understand right now and I would appreciate if the authors can help me understand the rest of it. Moreover, I would strongly encourage the authors to provide a high detailed technical summary of their contributions in the introduction (more suggestions in a later section).
- Section 3.1: Lemma 3.1 provides the conditions that need to be met for the optimal solution of eq. 4 and eq. 5. Proposition 3.1 provides what follows if those conditions are met by a solution under the assumption on the model's functional form. Then, I do not fully understand the claims around Remark 3.1. Does the problem of no good vanilla VI solution only happens if there is data point that has the same $x$ and $y$ but different $\delta$? The whole discussion around the overlapping sample spaces was very confusing for me. Overall, what is the main takeaway from this?
- Section 3.2: Overall, the things moved fast in Section 3.2 without a lot of commentry on what is happening and why. Several variables are introduced with no explanation of how to take this information. In particular, how do I interpret results of theorem 3.2.1? Does this imply that we need to keep separate variational parameters for the censored and the uncensored data? The formulation of remark 3.2 is weird. Why are we using $2-i$ here?
- Section 4.2: How is the $S$ implemented? Do we assume use of simple distributions where $S$ is available in closed form? What is $\zeta$ below figure 3? What purpose is proposition 4.2 serving in the training procedure?

---

> ### Author Rebuttal · Authors · 2025-03-27
>
> We truly appreciate your thoughtful and constructive feedback and your "willing to accept" recommendation. Your reasoning between what is well-explained and what is unclear is greatly appreciated. The following responses address the points you raised.
> # Evaluation: Best metrics, average metrics, more experiments.
>
> ### This is a general reponse. We value all reviewers’ thoughtful feedback on experimental design. While reporting best performance is a standard practice for VAEs (Burda et al. 2015; Tomczak & Welling 2018, Apellaniz et al. 2024), we agree that additional experiments are helpful. 4 additional experiments can be found via this [anonymous link](https://imgur.com/a/71JKyHg) with Apellaniz et al. 2024 included. Strong results across several benchmarks show our method consistently outperforms baselines on both best and average metrics. Meaningful exploration on simulated datasets is provided too.
> >Claims[1]: Question on vanilla VI solution, discussion around Remark 3.1, and its main takeaway
>
> Your interpretation of the vanilla VI failure case is spot on. Assuming the survival dataset contains two datapoints (x,y,1) and (x,y,0), optimal vanilla VI solution requires both inequalities (4) and (5) to hold as equalities at the same point (x,y) to perfect bound log-likelihood Eq.2. This leads to problems in Prop. 3.1. Remark 3.1 extends the VI optimality analysis to the population level by considering overlapping sample spaces. Specifically, it shows that, under non-informative censoring, an global optimal vanilla VI solution may fail to exist for non-degenerate θ (claim 1-2)  or may be trivial (claim 3-4).
> >Claims[2]: Section 3.2 is too fast. Interpreting Thm 3.2.1.
>
> Thank you for the comments. We conduct a standard VI optimality analysis (Domke & Sheldon, 2018) for log-likelihood Eq.2 in Section 3.2. Thm 3.2.1 formalizes the structure of the optimal variational posterior $q_ϕ$ that achieves a zero inference gap under general censoring assumptions. It shows that the optimal parameter ϕ inherently depends on both the event and censoring time distributions ($U$ and $C$). In addition, under certain censoring assumptions, its dependence on censoring time distribution $C$ can be *eliminated*, offering practical insights into model design.
> >Does this imply separating variational parameters for the censored/uncensored data?
>
> A general answer is no. A similar misconception was raised by Reviewer KuLt (#3). The encoder in Eq. 10 is a dense/joint network as shown in Fig3(a). Conditioning on 𝛿 = 0,1, it yields two branches within the same network, producing variational parameters $\phi_1$ and $\phi_2$, which are used for notational consistency, as noted in Line 203, with vanilla VI setup and Thm 4.3.1 to 4.3.3. That said, our codebase includes a *de facto* split encoder (`Delta_encoder`) that disables parameter sharing, though it should be used with discretion. For clarity, we have added an appendix section discussing why a joint encoder is theoretically preferred, as the two branches are inherently coupled under the VI optimality conditions.
> > Why $2−i$ is used in Remark 3.2?
>
> $2−i$ is used to unify the indices in the results. For $i=2, q(z|x,y) =q_{\phi_2}$ if and only if there is no event observation $p(\delta=2-2=0|y)=1$.
> >Claims[3]: How is the S implemented? Is S simple distributions?
>
> This likely refers to S(u∣x) in Eq.12. We compute S(u∣x) via the standard reparameterization trick by sampling z from the prior distribution and then computing closed-form $S(u|x,z) using the learned decoder (See `.predict()` in `model/cd-cvae.py`). While the decoder $f(u∣x,z)$ follows simple parametric forms (and so does S(u|x,z)), $f(u|x)$ and $S(u∣x)$ in Eq.2 are generally intractable, as they correspond to an infinite mixture over z [Rasmussen, 1999].
> >What is ζ below fig.3?
>
> For notation clarity, we decompose the decoder parameter θ into a location parameter $μ_ζ$ and a scale parameter σ. Specifically, ζ parameterizes the neural network that outputs the decoder mean $\mu(x, z)$.
> >What purpose is proposition 4.2 (no close-form update of $\sigma$)?
>
> As noted by Reviewer oybt (#2), stable training in VAE is non-trivial. While robust training strategies for VAEs are well-established in general settings (See Related Work, Liu & Wang, 2025), Proposition 4.2 identifies a fundamental pitfall: censored likelihoods in survival analysis make the dual-step optimization (Rybkin et al., 2021) *inapplicable*, justifying our use of standard $\sigma$ training in our proposed method of CDVI.
> > a motivating example/a detailed technical summary/a main take-away/add multiple references;
>
> We appreciate the valuable suggestions. Due to rebuttal length constraints, we will provide a detailed summary, supporting references, and an illustrative figure during the discussion phase.
>
> >  Suggestions [1-6]
>
> We appreciate the valuable comments and have addressed the noted minor issues. For clarification, DVI is our proposed delta method variant.

---

### Decision · Program_Chairs · 2025-05-01

**Decision:**

Accept (poster)

**Comment:**

After the discussion period, the reviewers unanimously recommended a "weak accept". It seems like the authors have largely addressed reviewer concerns although reviewers also pointed out there being clarity issues with the paper that seem to be hard to check during this reviewing process as to whether these issues are fixed (to be clear, this is not an issue with the reviewers not having enough background or lacking expertise; I think here it's on the authors to make their paper substantially clearer in exposition and motivation). I am recommending a weak accept for this paper as a result.